

# Evaluation of statistical methods for quantifying fractal scaling in water quality time series with irregular sampling

Qian Zhang[1], Ciaran J. Harman[2], James W. Kirchner[3,4,5]

[1] University of Maryland Center for Environmental Science at the US Environmental Protection Agency Chesapeake Bay Program Office, 410 Severn Avenue, Suite 112, Annapolis, MD 21403 (formerly, Department of Geography and Environmental Engineering, Johns Hopkins University, 3400 North Charles Street, Baltimore, Maryland 21218)

[2] Department of Environmental Health and Engineering, Johns Hopkins University, 3400 North Charles Street, Baltimore, Maryland 21218

[3] Department of Environmental System Sciences, ETH Zurich, Universitätstrasse 16, CH-8092 Zurich, Switzerland

[4] Swiss Federal Research Institute WSL, Zürcherstrasse 111, CH-8903 Birmensdorf, Switzerland

[5] Department of Earth and Planetary Science, University of California, Berkeley, California 94720

*Correspondence to*: Qian Zhang (qzhang@chesapeakebay.net)

**Abstract.** River water-quality time series often exhibit fractal scaling, which here refers to
autocorrelation that decays as a power law over some range of scales. Fractal scaling presents
challenges to the identification of deterministic trends, but traditional methods for estimating
spectral slope ($\beta$) or other equivalent scaling parameters (*e.g.*, Hurst exponent) are generally
inapplicable to irregularly sampled data. Here we consider two types of estimation approaches
for irregularly sampled data and evaluate their performance using synthetic time series. These
time series were generated such that (1) they exhibit a wide range of prescribed fractal scaling
behaviors, ranging from white noise ($\beta = 0$) to Brown noise ($\beta = 2$), and (2) their sampling gap
intervals mimic the sampling irregularity (as quantified by both the skewness and mean of gap-
interval lengths) in real water-quality data. The results suggest that none of the existing methods
fully account for the effects of sampling irregularity on $\beta$ estimation. First, the results illustrate
the danger of using interpolation for gap filling when examining auto-correlation, as the
interpolation methods consistently under-estimate or over-estimate $\beta$ under a wide range of
prescribed $\beta$ values and gap distributions. Second, the long-established Lomb-Scargle spectral
method also consistently under-estimates $\beta$. A modified form, using only the lowest 5% of the





frequencies for spectral slope estimation, has very poor precision, although the overall bias is
small. Third, a recent wavelet-based method, coupled with an aliasing filter, generally has the
smallest bias and root-mean-squared error among all methods for a wide range of prescribed $\beta$
values and gap distributions. The aliasing method, however, does not itself account for sampling
irregularity, and this introduces some bias in the result. Nonetheless, the wavelet method is
recommended for estimating $\beta$ in irregular time series until improved methods are developed.
Finally, all methods' performances depend strongly on the sampling irregularity, highlighting
that the accuracy and precision of each method are data-specific. Accurately quantifying the
strength of fractal scaling in irregular water-quality time series remains an unresolved challenge
for the hydrologic community and for other disciplines that must grapple with irregular sampling.
**Key Words**
Fractal scaling, autocorrelation, Hurst effect, river water-quality sampling, sampling irregularity,
trend analysis
**1. Introduction**
*1.1. Autocorrelations in Time Series*

It is well known that time series from natural systems often exhibit auto-correlation, that is,

observations at each time step are correlated with observations one or more time steps in the past.
This property is usually characterized by the autocorrelation function (ACF), which is defined as
follows for a process $X_t$ at lag $k$:

$$\gamma(k) = cov(X_t, X_{t+k}) \qquad (1)$$

In practice, auto-correlation has been frequently modeled with classical techniques such as auto-
regressive (AR) or auto-regressive moving-average (ARMA) models (Darken *et al.*, 2002; Yue
*et al.*, 2002; Box *et al.*, 2008). These models assume that the underlying process has short-term
memory, *i.e.*, the ACF decays exponentially with lag $k$, which implies that the ACF is summable
(Box *et al.*, 2008).

Although the short-term memory assumption holds sometimes, it cannot adequately describe

many time series whose ACFs decay as a power law (thus much slower than exponentially) and
may not reach zero even for large lags, which implies that the ACF is non-summable. This





property is commonly referred to as long-term memory or fractal scaling, as opposed to short-
term memory (Beran, 2010).

### 1.2. Overview of Approaches for Quantification of Fractal Scaling

Several equivalent metrics can be used to quantify fractal scaling. Here we provide a review
of the definitions of such processes and several typical modeling approaches, including both
time-domain and frequency-domain techniques, with special attention to their reconciliation. For
a more comprehensive review, readers are referred to Beran *et al.* (2013), Boutahar *et al.* (2007),
and Witt and Malamud (2013).
Strictly speaking, $X_t$ is called a stationary long-memory process if the condition

$$\lim_{k \to \infty} k^\alpha \gamma(k) = C_1 > 0 \qquad (2)$$

where $C_1$ is a constant, is satisfied by some $\alpha \in (0,1)$ (Boutahar *et al.*, 2007; Beran *et al.*, 2013).
Equivalently, $X_t$ is a long-memory process if, in the spectral domain, the condition

$$\lim_{\omega \to 0} |\omega|^\beta f(\omega) = C_2 > 0 \qquad (3)$$

is satisfied by some $\beta \in (0,1)$, where $C_2$ is a constant and $f(\omega)$ is the spectral density function
of $X_t$, which is related to ACF as follows (which is also known as the Wiener-Khinchin theorem):

$$f(\omega) = \frac{1}{2\pi} \sum_{k=-\infty}^{\infty} \gamma(k) e^{-ik\omega} \qquad (4)$$

where $\omega$ is angular frequency (Boutahar *et al.*, 2007).
One popular model for describing long-memory processes is the so-called fractional auto-
regressive integrated moving-average model, or ARFIMA ($p$, $q$, $d$), which is an extension of
ARMA models and is defined as follows:

$$(1 - B)^d \varphi(B) X_t = \psi(B) \varepsilon_t \qquad (5)$$

where $\varepsilon_t$ is a series of independent, identically distributed Gaussian random numbers $\sim (0, \sigma_\varepsilon^2)$,
$B$ is the backshift operator (*i.e.*, $BX_t = X_{t-1}$), and functions $\varphi(\cdot)$ and $\psi(\cdot)$ are polynomials of order
$p$ and $q$, respectively. The fractional differencing parameter $d$ is related to the parameter $\alpha$ in Eq.
(2) as follows:

$$d = \frac{1 - \alpha}{2} \in (-0.5, 0.5) \qquad (6)$$

(Beran *et al.*, 2013; Witt and Malamud, 2013).



65   In addition to a slowly decaying ACF, a long-memory process manifests itself in two other

66 equivalent fashions. One is the so-called "Hurst effect", which states that, on a log-log scale, the

67 range of variability of a process changes linearly with the length of time period under

68 consideration. This power-law slope is often referred to as the "Hurst exponent" or "Hurst

69 coefficient" $H$ (Hurst, 1951), which is related to $d$ as follows:

$$H = d + 0.5 \tag{7}$$

70 (Beran *et al.*, 2013; Witt and Malamud, 2013). The second equivalent description of long-

71 memory processes, this time from a frequency-domain perspective, is "fractal scaling", which

72 describes a power-law decrease in spectral power with increasing frequency, yielding power

73 spectra that are linear on log-log axes (Lomb, 1976; Scargle, 1982; Kirchner, 2005).

74 Mathematically, this inverse proportionality can be expressed as:

$$f(\omega) = C_3 |\omega|^{-\beta} \tag{8}$$

75 where $C_3$ is a constant and the scaling exponent $\beta$ is termed the "spectral slope." In particular, for

76 spectral slopes of zero, one, and two, the underlying processes are termed as "white", "pink" (or

77 "flicker"), and "Brown" (or "red") noises, respectively (Witt and Malamud, 2013). Illustrative

78 examples of these three noises are shown in **Figure 1a-1c**.

79   In addition, it can be shown that the spectral density function for ARFIMA ($p,d,q$) is

$$f(\omega) = \frac{\sigma_\varepsilon^2}{2\pi} \frac{|\psi(e^{-i\omega})|^2}{|\varphi(e^{-i\omega})|^2} |1 - e^{-i\omega}|^{-2d} \tag{9}$$

80 for $-\pi < \omega < \pi$ (Boutahar *et al.*, 2007; Beran *et al.*, 2013). For $|\omega| \ll 1$, Eq. (9) can be

81 approximated by:

$$f(\omega) = C_4 |\omega|^{-2d} \tag{10}$$

82 with

$$C_4 = \frac{\sigma_\varepsilon^2}{2\pi} \frac{|\psi(1)|^2}{|\varphi(1)|^2} \tag{11}$$

83 Eq. (10) thus exhibits the asymptotic behavior required for a long-memory process given by Eq.

84 (3). In addition, a comparison of Eq. (10) and (8) reveals that,

$$\beta = 2d \tag{12}$$

85 Overall, these derivations indicate that these different types of scaling parameters (*i.e.*, $\alpha$, $d$, and

86 $H$ and $\beta$) can be used equivalently to describe the strength of fractal scaling. Specifically, their

87 equivalency can be summarized as follows:



$$\beta = 2d = 1 - \alpha = 2H - 1 \qquad (13)$$

It should be noted, however, that the parameters $d$, $\alpha$, and $H$ are only applicable over a fixed
range of fractal scaling, which is equivalent to (-1, 1) in terms of $\beta$.

Fractal scaling has been increasingly recognized in studies of hydrological time series,

particularly for the common task of trend identification. Such hydrological series include
riverflow (Montanari *et al.*, 2000; Khaliq *et al.*, 2008; Khaliq *et al.*, 2009; Ehsanzadeh and
Adamowski, 2010), air and sea temperature (Fatichi *et al.*, 2009; Lennartz and Bunde, 2009;
Franzke, 2012b; Franzke, 2012a), conservative tracers (Kirchner *et al.*, 2000; Kirchner *et al.*,
2001; Godsey *et al.*, 2010), and non-conservative chemical constituents (Kirchner and Neal,
2013; Aubert *et al.*, 2014). Because for fractal scaling processes the variance of the sample mean
converges to zero much slower than the rate of $n^{-1}$ (n: sample size), the fractal scaling property
must be taken into account to avoid "false positives" (Type I errors) when inferring the statistical
significance of trends (Cohn and Lins, 2005; Fatichi *et al.*, 2009; Ehsanzadeh and Adamowski,
2010; Franzke, 2012a). Unfortunately, as stressed by Cohn and Lins (2005), it is "surprising that
nearly every assessment of trend significance in geophysical variables published during the past
few decades has failed [to do so]", and a similar tendency is evident in the decade following that
statement as well.
*1.3. Motivation and Objective of this Work*

To account for fractal scaling in trend analysis, one must be able to first quantify the strength

of fractal scaling for a given time series. Numerous estimation methods have been developed for
this purpose, including Hurst rescaled range analysis, Higuchi's method, Geweke and Porter-
Hudak's method, Whittle's maximum likelihood estimator, detrended fluctuation analysis, and
others (Taqqu *et al.*, 1995; Montanari *et al.*, 1997; Montanari *et al.*, 1999; Rea *et al.*, 2009;
Stroe-Kunold *et al.*, 2009). For brevity, these methods are not elaborated here; readers are
referred to Beran (2010) and Witt and Malamud (2013) for details. While these estimation
methods have been extensively adopted, they are unfortunately only applicable to regular (*i.e.*,
evenly spaced) data, *e.g.*, daily streamflow discharge, monthly temperature, *etc*. In practice,
many types of hydrological data, including river water-quality data, are often sampled irregularly
or have missing values, and hence their strengths of fractal scaling cannot be readily estimated
with the above traditional estimation methods.



Thus, estimation of fractal scaling in irregularly sampled data is an important challenge for
hydrologists and practitioners. Many data analysts may be tempted to interpolate the time series
to make it regular and hence analyzable (Graham, 2009). Although technically convenient,
interpolation can be problematic if it distorts the series' autocorrelation structure (Kirchner and
Weil, 1998). In this regard, it is important to evaluate various types of interpolation methods
using carefully designed benchmark tests and to identify the scenarios under which the
interpolated data can yield reliable (or, alternatively, biased) estimates of spectral slope.
Moreover, quantification of fractal scaling in real-world water-quality data is subject to
several common complexities. First, water-quality data are rarely normally distributed; instead,
they are typically characterized by log-normal or other skewed distributions (Hirsch *et al.*, 1991;
Helsel and Hirsch, 2002), with potential consequences for $\beta$ estimation. Moreover, water-quality
data also tend to exhibit long-term trends, seasonality, and flow-dependence (Hirsch *et al.*, 1991;
Helsel and Hirsch, 2002), which can also affect the accuracy of $\beta$ estimate. Thus, it may be more
plausible to quantify $\beta$ in transformed time series after accounting for the seasonal patterns and
discharge-driven variations in the original time series, which is also the approach taken in this
work. For the trend aspect, however, it remains a puzzle whether the data set should be de-
trended before conducting $\beta$ estimation. Such de-trending treatment can certainly affect the
estimated value of $\beta$ and hence the validity of (or confidence in) any inference made regarding
the statistical significance of temporal trends in the time series. This somewhat circular issue is
beyond the scope of our current work -- it has been previously discussed in the context of short-
term memory (Zetterqvist, 1991; Darken *et al.*, 2002; Yue *et al.*, 2002; Noguchi *et al.*, 2011;
Clarke, 2013; Sang *et al.*, 2014), but it is not well understood in the context of fractal scaling (or
long-term memory) and hence presents an important area for future research.
In the above context, the main objective of this work was to use Monte Carlo simulation to
systematically evaluate and compare two broad types of approaches for estimating the strength
of fractal scaling (*i.e.*, spectral slope $\beta$) in irregularly sampled river water-quality time series.
Specific aims of this work include the following:
(1) To examine the sampling irregularity of typical river water-quality monitoring data and

to simulate time series that contain such irregularity; and

(2) To evaluate two broad types of approaches for estimating $\beta$ in simulated irregularly

sampled time series.





The first type of approach includes several forms of interpolation techniques for gap filling, thus
making the data regular and analyzable by traditional estimation methods. The second type of
approach includes the well-known Lomb-Scargle periodogram (Lomb, 1976; Scargle, 1982) and
a recently developed wavelet method combined with a spectral aliasing filter (Kirchner and Neal,
2013). The latter two methods can be directly applied to irregularly spaced data; here we aim to
compare them with the interpolation techniques. Details of these various approaches are
provided in **Section 3.1**.

This work was designed to make several specific contributions. First, it uses benchmark tests

to quantify the performance of a wide range of methods for estimating fractal scaling in
irregularly sampled water-quality data. Second, it proposes an innovative and general approach
for modeling sampling irregularity in water-quality records. Third, while this work was not
intended to compare all published estimation methods for fractal scaling, it does provide and
demonstrate a generalizable framework for data simulation (with gaps) and $\beta$ estimation, which
can be readily applied toward the evaluation of other methods that are not covered here. Last but
not least, while this work was intended to help hydrologists and practitioners understand the
performance of various approaches for water-quality time series, the findings and approaches
may be broadly applicable to irregularly sampled data in many other scientific disciplines.

The rest of the paper is organized as follows. We propose a general approach for modeling

sampling irregularity in typical river water-quality data and discuss our approach for simulating
irregularly sampled data (**Section 2**). We then introduce the various methods for estimating
fractal scaling in irregular time series and compare their estimation performance (**Section 3**). We
close with a discussion of the results and implications (**Section 4**).
**2. Quantification of Sampling Irregularity in River Water-Quality Data**
*2.1. Modeling of Sampling Irregularity*

River water-quality data are often sampled irregularly. In some cases, samples are taken

more frequently during particular periods of interest, such as high flows or drought periods; here
we will address the implications of the irregularity, but not the (intentional) bias, inherent in such
a sampling strategy. In other cases, the sampling is planned with a fixed sampling interval (*e.g.*,
1 day) but samples are missed (or lost, or fail quality-control checks) at some time steps during
implementation. In still other cases, the sampling is intrinsically irregular because, for example,





one cannot measure the chemistry of rainfall on rainless days or the chemistry of a stream that
has dried up. Theoretically, any deviation from fixed-interval sampling can affect the subsequent
analysis of the time series.
To quantify the sampling irregularity, we propose a simple and general approach that can be
applied to any time series of monitoring data. Specifically, for a given time series with $N$ points,
the time intervals between adjacent samples are calculated; these intervals themselves make up a
time series of $N$-1 points that we call $\Delta t$. In addition, the following parameters are calculated to
quantify its sampling irregularity:
•   $L$ = the length of the period of record,
•   $N$ = the number of samples in the record,
•   $\Delta t_{nominal}$ = the nominal sampling interval under regular sampling (*e.g.*, $\Delta t_{nominal}$ = 1 day

for daily samples),

•   $\Delta t^* = \Delta t / \Delta t_{nominal}$, the sample intervals non-dimensionalized by the nominal sampling

interval,

•   $\Delta t_{average} = L/(N - 1)$ the average of all the entries in $\Delta t$.
The quantification is illustrated with two simple examples. The first example contains data
sampled every hour from 1:00 am to 11:00 am on one day. In this case, $L$ = 10 hours, $N$ = 11
samples, $\Delta t$ = {1, 1, 1, 1, 1, 1, 1, 1, 1, 1} hour, and $\Delta t_{nominal} = \Delta t_{average}$ = 1 hour. The second
example contains data sampled at 1:00 am, 3:00 am, 4:00 am, 8:00 am, and 11:00 am. In this
case, $L$ = 10 hours, $N$ = 5 samples, $\Delta t$ = {2, 1, 4, 3} hours, $\Delta t_{nominal}$ = 1 hour, and $\Delta t_{average}$ = 2.5
hours. It is readily evident that the first case corresponds to fixed-interval (regular) sampling that
has the property of $\Delta t_{average}/\Delta t_{nominal}$ = 1 (dimensionless), whereas the second case corresponds to
irregular sampling for which $\Delta t_{average}/\Delta t_{nominal} > 1$.
The dimensionless set $\Delta t^*$ contains essential information for determining sampling
irregularity. This set is modeled as independent, identically distributed values drawn from a
negative binomial (NB) distribution. This distribution has two dimensionless parameters, the
shape parameter ($\lambda$) and the mean parameter ($\mu$), which collectively represent the irregularity of
the samples. The NB distribution is a flexible distribution that provides a discrete analogue of a
gamma distribution. The geometric distribution, itself the discrete analogue of the exponential
distribution, is a special case of the NB distribution when $\lambda$ = 1.





The parameters $\mu$ and $\lambda$ represent different aspects of sampling irregularity, as illustrated by
the examples shown in **Figure 2**. The mean parameter $\mu$ represents the fractional increase in the
average interval between samples due to gaps: $\mu = \text{mean}(\Delta t^*) - 1 = (\Delta t_{average} - \Delta t_{nominal})/\Delta t_{nominal}$.
Thus the special case of $\mu = 0$ corresponds to regular sampling (*i.e.*, $\Delta t_{average} = \Delta t_{nominal}$), whereas
any larger value of $\mu$ corresponds to irregular sampling (*i.e.*, $\Delta t_{average} > \Delta t_{nominal}$) (**Figure 2c**). The
shape parameter $\lambda$ characterizes the similarity of gaps to each other; that is, a small $\lambda$ indicates
that the samples contain gaps of widely varying lengths, whereas a large $\lambda$ indicates that the
samples contain many gaps of similar lengths (**Figure 2a-2b**).
To visually illustrate these gap distributions, representative samples of gappy time series are
presented in **Figure 1** for the three special processes described above (**Section 1.2**), *i.e.*, white
noise, pink noise, and Brown noise. Specifically, three different gap distributions, namely, NB($\lambda$
$= 1, \mu = 1$), NB($\lambda = 1, \mu = 14$), and NB($\lambda = 0.01, \mu = 1$), were simulated and each was applied to
convert the three original (regular) time series (**Figure 1a-1c**) to gappy time series (**Figure 1d-**
**1l**). These simulations clearly illustrate the effects of the two parameters $\lambda$ and $\mu$. In particular,
compared with NB($\lambda = 1, \mu = 1$), NB($\lambda = 1, \mu = 14$) shows a similar level of sampling irregularity
(same $\lambda$) but a much longer average gap interval (larger $\mu$). Again compared with NB($\lambda = 1, \mu =$
1), NB($\lambda = 0.01, \mu = 1$) shows the same average interval (same $\mu$) but a much more irregular
(skewed) gap distribution that contains a few very large gaps (smaller $\lambda$).
### *2.2. Examination of Sampling Irregularity in Real River Water-Quality Data*
The above modeling approach was applied to real water-quality data from two large river
monitoring networks in the United States to examine sampling irregularity. One such network is
the Chesapeake Bay River Input Monitoring program, which typically samples streams bi-
monthly to monthly, accompanied with additional sampling during stormflows (Langland *et al.*,
2012; Zhang *et al.*, 2015). These data were obtained from the U.S. Geological Survey National
Water Information System (http://doi.org/10.5066/F7P55KJN). The other network is the Lake
Erie and Ohio tributary monitoring program, which typically samples streams at a daily
resolution (National Center for Water Quality Research, 2015). For each site, we determined the
NB parameters to quantify sampling irregularity. The mean parameter $\mu$ can be estimated as
described above, and the shape parameter $\lambda$ can be calculated directly from the mean and
variance of $\Delta t^*$ as follows: $\lambda = \mu^2/[\text{var}(\Delta t^*) - \mu] = (\text{mean}(\Delta t^*) - 1)^2/[\text{var}(\Delta t^*) - \text{mean}(\Delta t^*) + 1]$.
Alternatively, a maximum likelihood approach can be used, which employs the "*fitdist*" function





in the "*fitdistrplus*" R package (Delignette-Muller and Dutang, 2015). In general, the two
approaches have produced similar results, which are summarized in **Table 1**, with two examples
of fitted NB distributions shown in **Figure 3**.

For the Chesapeake Bay River Input Monitoring program (9 sites), total nitrogen (TN) and

total phosphorus (TP) are taken as representatives of water-quality constituents. According to the
maximum likelihood approach, the shape parameter $\lambda$ varies between 0.7 and 1.2 for TN and
between 0.8 and 1.1 for TP (**Table 1**). These $\lambda$ values are around 1.0, reflecting the fact that
these sites have relatively even gap distributions (*i.e.*, relatively balanced counts of large and
small gaps). The mean parameter $\mu$ varies between 9.5 and 19.6 for TN and between 13.4 and
24.4 for TP in the Chesapeake monitoring network, corresponding to $\Delta t_{average}$ of 10.5–20.6 days
for TN and 14.4–25.4 days for TP, respectively. This is consistent with the fact that these sites
have typically been sampled bi-monthly to monthly, along with additional sampling during
stormflows (Langland *et al.*, 2012; Zhang *et al.*, 2015).

For the Lake Erie and Ohio tributary monitoring program (6 sites), the record of nitrate-plus-

nitrite ($NO_x$) and TP were examined. According to the maximum likelihood approach, the shape
parameter $\lambda$ is approximately 0.01 for both constituents (**Table 1**). These very low $\lambda$ values occur
because these time series contain a few very large gaps, ranging from 35 days to 1109 days (~3
years). The mean parameter $\mu$ varies between 0.06 and 0.22, corresponding to $\Delta t_{average}$ of 1.06
and 1.22 days, respectively. This is consistent with fact that these sites have been sampled at a
daily resolution with occasional missing values on some days (Zhang and Ball, 2017).
*2.3. Simulation of Time Series with Irregular Sampling*

To evaluate the various $\beta$ estimation methods, our first step was to use Monte Carlo

simulation to produce time series that mimic the sampling irregularity observed in real water-
quality monitoring data. We began by simulating regular (gap free) time series using the
fractional noise simulation method of Witt and Malamud (2013), which is based on inverse
Fourier filtering of white noises. Our analysis showed this method performed well compared to
other simulation methods for $\beta$ values between 0 and 1 (data not shown). In addition, this
method can also simulate $\beta$ values beyond this range. The noises simulated by the Witt and
Malamud method, however, are band-limited to the Nyquist frequency (half of the sampling
frequency) of the underlying white noise time series, whereas true fractional noises would
contain spectral power at all frequencies, extending well above the Nyquist frequency for any





sampling. Thus these band-limited noises will be less susceptible to spectral aliasing than true
fractional noises would be; see Kirchner (2005) for detailed discussions of the aliasing issue.

Thirty replicates of regular (gap free) time series were produced for nine prescribed spectral

slopes, which vary from $\beta = 0$ (white noise) to $\beta = 2$ (Brownian motion or "random walk") with
an increment of 0.25 (*i.e.*, 0, 0.25, 0.5, 0.75, 1.0, 1.25, 1.5, 1.75, and 2). These regular time series
each have a length ($N$) of 9125, which can be interpreted as 25 years of regular daily samples
(that is, $\Delta t_{nominal} = 1$ day).

Each of the simulated regular time series was converted to irregular time series using gap

intervals that were simulated with NB distributions. To make these gap intervals mimic those in
typical river water-quality time series, representative NB parameters were chosen based on
results from **Section 2.2**. Specifically, $\mu$ was set at 1 and 14, corresponding to $\Delta t_{average}$ of 2 days
and 15 days respectively. For $\lambda$, we chose four values that span three orders of magnitude, *i.e.*,
0.001, 0.1, 1, and 10. Note that when $\lambda = 1$ the generated time series corresponds to a Bernoulli
process. With the chosen values of $\mu$ and $\lambda$, a total of eight scenarios were generated, which were
implemented using the "*rnbinom*" function in the "*stats*" R package (R Development Core Team,

2014):

1)  $\mu = 1$ (*i.e.*, $\Delta t_{average}/\Delta t_{nominal} = 2$), $\lambda = 0.01$,

2)  $\mu = 1$, $\lambda = 0.1$,

3)  $\mu = 1$, $\lambda = 1$,

4)  $\mu = 1$, $\lambda = 10$,

5)  $\mu = 14$ (*i.e.*, $\Delta t_{average}/\Delta t_{nominal} = 15$), $\lambda = 0.01$,

6)  $\mu = 14$, $\lambda = 0.1$,

7)  $\mu = 14$, $\lambda = 1$,

8)  $\mu = 14$, $\lambda = 10$.

Examples of these simulations are shown with boxplots in **Figure 2**.
**3. Evaluation of Proposed Estimation Methods for Irregular Time Series**
*3.1. Summary of Estimation Methods*

For the simulated irregular time series, $\beta$ was estimated using the aforementioned two types

of approaches. The first type includes 11 different interpolation methods (designated as B1-B11
below) to fill the data gaps, thus making the data regular and analyzable by traditional methods:



B1)    Global mean: all missing values replaced with the mean of all observations;
B2)    Global median: all missing values replaced with the median of all observations;
B3)    Random replacement: all missing values replaced with observations randomly drawn

(with replacement) from the time series;

B4)    Next observation carried backward: each missing value replaced with the next available

observation;

B5)    Last observation carried forward: each missing value replaced with the preceding

available observation;

B6)    Average of the two nearest samples: it replaces each missing value with the mean of its

next and preceding available observations;

B7)    Lowess (locally weighted scatterplot smoothing) with a smoothing span of 1: missing

values replaced using fitted values from a lowess model determined using all available

observations (Cleveland, 1981);

B8)    Lowess with a smoothing span of 0.75: same as B7 except that the smoothing span is 75%

of the available data (similar distinction follows for B9-B11);

B9)    Lowess with a smoothing span of 50%;
B10)   Lowess with a smoothing span of 30%; and
B11)   Lowess with a smoothing span of 10%.
B4 and B5 were implemented using the "*na.locf*" function in the "*zoo*" R package (Zeileis and
Grothendieck, 2005). B7-B11 were implemented using the "*loess*" function in the "*stats*" R
package (R Development Core Team, 2014). An illustration of these interpolation methods is
provided in **Figure 4**. The interpolated data, along with the original regular data (designated as
A1) were analyzed using the Whittle's maximum likelihood method for $\beta$ estimation, which was
implemented using the "*FDWhittle*" function in the "*fractal*" R package (Constantine and
Percival, 2014).
The second type of approaches estimates $\beta$ in the irregularly sampled data directly, using
several variants of the Lomb-Scargle periodogram (designated as C1a-C1c below), and a
recently developed wavelet-based method (designated as C2 below). Specifically, these
approaches are:
C1a)   Lomb-Scargle periodogram: the spectral density of the time series (with gaps) is

estimated and the spectral slope is fit using all frequencies (Lomb, 1976; Scargle, 1982).



This is a classic method for examining periodicity in irregularly sampled data, which is

analogous to the more familiar fast Fourier transform method often used for regularly

sampled data;

C1b) Lomb-Scargle periodogram with 5% data: same as C1a except that the fitting of the

spectral slope considers only the lowest 5% frequencies (Montanari *et al.*, 1999);

C1c) Lomb-Scargle periodogram with "binned" data: same as C1a except that the fitting of

the spectral slope is performed on binned data in three steps: (1) The entire range of

frequency is divided into 100 equal-interval bins on logarithmic scale. (2) The

respective medians of frequency and power spectral density are calculated for each of

the 100 bins. (3) The 100 pairs of median frequency and median spectral density are

used to estimate the spectral slope on a log-log scale.

C2)  Kirchner and Neal (2013)'s wavelet method: uses a modified version of Foster's

weighted wavelet spectrum (Foster, 1996) to suppress spectral leakage from low

frequencies and applies an aliasing filter (Kirchner, 2005) to remove spectral aliasing

artifacts at high frequencies.

C1a was implemented using the "*spec.ls*" function in the "*cts*" R package (Wang, 2013). C2 was
run in *C*, using codes modified from those in Kirchner and Neal (2013).
***3.2. Evaluation of Methods' Performance***
Each estimation method listed above was applied to the simulated data (**Section 2.3**) to
estimate $\beta$, which were then compared with the prescribed ("true") $\beta$ to quantify the performance
of each method. Plots of method evaluation for all simulations are provided as **Figures S1-S10**
in the Supporting Information. Close inspections of these plots reveal some general patterns of
the methods' performance. For brevity, these patterns are presented with a subset of the plots,
which correspond to the cases where true $\beta = 1$ and shape parameter $\lambda = 0.01, 0.1, 1,$ and 10
(**Figure 5**). In general, $\beta$ values estimated using the regular data (A1) are very close to 1.0,
which indicates that the adopted fractional noise generation method and the Whittle's maximum
likelihood estimator have small combined simulation and estimation bias. This is perhaps
unsurprising, since the estimator is based on the Fourier transform and the noise generator is
based on an inverse Fourier transform; thus, one method is essentially just the inverse of the
other. One should also note that when fractional noises are not arbitrarily band-limited at the
Nyquist frequency (as they inherently are with the noise generator that is used here), spectral



aliasing should lead to spectral slopes that are flatter than expected (Kirchner, 2005), and thus to
underestimates of LRD.
For the simulated irregular data, the estimation methods differ widely in their performance.
Specifically, three interpolation methods (*i.e.*, B4-B6) consistently over-estimate $\beta$, indicating
that they introduce additional correlations into the time series, reducing its short-timescale
variability. In contrast, the other eight interpolation methods (*i.e.*, B1-B3 and B7-B11) generally
under-estimate $\beta$, indicating that the interpolated points are less correlated than the original time
series, thus introducing additional variability on short timescales. As expected, results from the
lowess methods (B7-B11) depend strongly on the size of smoothing window, that is, more
severe under-estimation of $\beta$ is produced as the smoothing window becomes wider. In fact, when
the smoothing window is 1.0 (*i.e.*, method B7), lowess performs the interpolation using all data
available and thus behaves similarly to interpolations based on global means (B1) or global
medians (B2), except that lowess fits a polynomial curve instead of constant values. However,
whenever a sampling gap is much shorter than the smoothing window, the infilled lowess value
will be close to the local mean or median, and the abrupt jumps produced by these infilled values
will artificially increase the variance in the time series at high frequencies, leading to an
artificially reduced spectral slope $\beta$ and correspondingly, an underestimate of $\beta$. This mechanism
explains why lowess interpolation distorts $\beta$ more when there are many small gaps (large $\lambda$), and
therefore more jumps to, and away from, the infilled values, than when there are only a few large
gaps (small $\lambda$).
Among the direct methods (*i.e.*, C1a, C1b, C1c, and C2), the Lomb-Scargle method, with
original data (C1a) or binned data (C1c) tends to under-estimate $\beta$, though the underestimation
by C1c is generally less severe. The modified Lomb-Scargle method (C1b), using only the
lowest 5% of frequencies, yields estimates that are centered around 1.0. However, C1b has the
highest variability (*i.e.*, least precision) in $\beta$ estimates among all methods. Compared with all the
above methods, the wavelet method (C2) has much better performance in terms of both accuracy
and precision when $\lambda$ is 1 or 10, a slightly better performance when $\lambda$ is 0.1, but a worse
performance when $\lambda$ is 0.01.
The shape parameter $\lambda$ greatly affects the performance of the estimation methods. All the
interpolation methods that under-estimate $\beta$ (*i.e.*, B1-B3 and B7-B11) perform worse as $\lambda$
increases from 0.01 to 10. This effect can be interpreted as follows: when the time series



contains a large number of relatively small gaps (*e.g.*, $\lambda = 1$ or 10), there are many jumps (which,
as noted above, contain mostly high-frequency variance) between the original data and the
infilled values, resulting in more severe under-estimation. In contrast, when the data contain only
a small number of very large gaps (*e.g.*, $\lambda = 0.01$ or 0.1), there are fewer of these jumps, resulting
in minimal under-estimation. Similar effects of $\lambda$ are also observed with the interpolation
methods that show over-estimation (*i.e.*, B4-B6) – that is, over-estimation is more severe when $\lambda$
is larger. Similarly, the Lomb-Scargle method (C1a and C1c) performs worse (more serious
underestimation) as $\lambda$ increases. Finally, method C2 seems to perform the best when $\lambda$ is large (1
or 10), but not well when $\lambda$ is very small (0.01), as noted above. This result highlights the
sensitivity of the wavelet method to the presence of a few large gaps in the time series. For such
cases, a potentially more feasible approach is to break the whole time series into several
segments (each without long gaps) and then apply the wavelet method (C2) to analyze each
segment separately. If this can yield more accurate estimates, then further simulation
experiments should be designed to systematically determine how long the gap needs to be to
invoke such an approach.

Next, the method evaluation is extended to all the simulated spectral slopes, that is, $\beta = 0$,

0.25, 0.5, 0.75, 1.0, 1.25, 1.5, 1.75, and 2. For ease of discussion, three quantitative criteria were
proposed for evaluating performance, namely, bias (B), standard deviation (SD), and root-mean-
squared error (RMSE), as defined below:

$$B_i = \overline{\beta}_i - \beta_{true} \tag{14}$$

$$SD_i = \sqrt{\frac{1}{29}\sum_{j=1}^{30}(\beta_{i,j} - \overline{\beta}_i)^2} \tag{15}$$

$$RMSE_i = \sqrt{B_i^2 + SD_i^2} \tag{16}$$

where $\overline{\beta}_i$ is the mean of 30 $\beta$ values estimated by method i, and $\beta_{true}$ is the prescribed $\beta$ value for
simulation of the initial regular time series. In general, B and SD can be considered as the
models' systematic error and random error, respectively, and RMSE serves as an integrated
measure of both errors. For all evaluations, plots of bias and RMSE are provided in the main text.
(Plots of SD are provided as **Figure S5** and **Figure S10** for simulations with $\mu = 1$ and $\mu = 14$,
respectively.)





For simulations with $\mu = 1$, results of estimation bias and RMSE are summarized in **Figure 6**
and **Figure 7**, respectively. (More details are provided in **Figures S1-S4**.) For brevity, we focus
on three direct methods (C1a, C1b and C2) and three representative interpolation methods.
(Specifically, B1 represents B1-B3 and B7; B6 represents B4-B6, and B8 represents B8-B11.)
Overall, these six methods show mixed performances. In terms of bias (**Figure 6**), B1 (global
mean) and B8 (lowess with a smoothing span of 0.75) tend to have negative bias, particularly for
time series with (1) moderate-to-large $\beta_{true}$ values and (2) large $\lambda$ values (*i.e.*, less skewed gap
intervals). By contrast, B1 and B8 generally have minimal bias when (1) $\beta_{true}$ is close to zero (*i.e.*,
when the simulated time series is close to white noise); and (2) $\lambda$ is small (*e.g.*, 0.01), since
interpolating a few large gaps cannot significantly affect the overall correlation structure. In
addition, lowess interpolation with a larger smoothing window tends to yield more negatively
biased estimates (data not shown). The other interpolation method, B6 (mean of the two nearest
neighbors) tends to over-estimate $\beta$, particularly for time series with (1) small $\beta_{true}$ values and (2)
large $\lambda$ values. At large $\beta_{true}$ values (*e.g.*, 2.0), the auto-correlation is already very strong such
that taking the mean of two neighbors for gap filling does not introduce much additional
correlation, as opposed to the case of small $\beta_{true}$ values. The Lomb-Scargle methods (C1a and
C1b) generally have negative bias, particularly for time series with (1) moderate-to-large $\beta_{true}$
values (for both methods) and (2) large $\lambda$ values (for C1a), which is similar to B1 and B8.
However, C1b overall shows less severe bias than C1a. Finally, the wavelet method (C2) shows
generally the smallest bias among all methods. However, its performance advantage is not as
great when the time series has small $\lambda$ values (*i.e.*, very skewed gap intervals), as noted above,
which may be due to the fact that the aliasing filter was designed for regular time series. In terms
of SD (**Figure S5**), method C1b performs the worst among all methods (as noted above), method
B6 and B8 perform poorly for large $\beta_{true}$ values, and method C2 performs poorly for $\beta_{true} = 0$. In
terms of RMSE (**Figure 7**), methods B1, B8, C1a, and C1b perform well for small $\beta_{true}$ values
and small $\lambda$ values, whereas method B6 performs well for large $\beta_{true}$ values and small $\lambda$ values. In
comparison, method C2 has the smallest RMSEs among all methods, and its RMSEs are
similarly small for the wide range of $\beta_{true}$ and $\lambda$ values. In general, the wavelet method can be
considered the best among all methods.
For simulations with $\mu = 14$, results of estimation bias and RMSE are summarized in
**Figure 8** and **Figure 9**, respectively. (More details are provided in **Figures S6-S9**.) Overall,





these methods show mixed performances that are generally similar to the cases when $\mu = 1$, as
discussed above. These results highlight the generality of these methods' performances, which
applies at least to the range of $\mu = [1, 14]$. In addition, all methods show generally larger RMSE
for $\mu = 14$ than $\mu = 1$, indicating their dependence on the mean gap interval (**Figure 9**). Perhaps
the most notable difference is observed with method C2, which in this case shows positive bias
for small $\lambda$ values (0.01 and 0.1) and negative bias for large $\lambda$ values (1 and 10) (**Figure 8f**). It
nonetheless generally shows the smallest RMSEs among all the tested methods.
*3.3. Quantification of Spectral Slopes in Real Water-Quality Data*
In this section, the proposed estimation approaches were applied to quantify $\beta$ in real water-
quality data from the two monitoring programs presented in **Section 2.2** (**Table 1**). As noted in
**Section 1.3**, such real data are typically much more complex than our simulated time series,
because of (1) strong deviations from normal distributions and (2) effects of flow-dependence,
seasonality, and temporal trend (Hirsch *et al.*, 1991; Helsel and Hirsch, 2002). In this regard,
future research may simulate time series with these important characteristics and evaluate the
performance of various estimation approaches, perhaps following the modeling framework
described herein. Alternatively, one may quantify $\beta$ in transformed time series after accounting
for the above aspects. In this work, we have taken the latter approach for a preliminary
investigation. Specifically, we have used the published Weighted Regressions on Time,
Discharge, and Season (WRTDS) method (Hirsch *et al.*, 2010) to transform the original time
series. This widely accepted method estimates daily concentrations based on discretely collected
concentration samples using time, season, and discharge as explanatory variables, *i.e.*,

$$ln(C) = \beta_0 + \beta_1 t + \beta_2 ln(Q) + \beta_3 sint(2\pi t) + \beta_4 cos(2\pi t) + \varepsilon \qquad (17)$$

where $C$ is concentration, $Q$ is daily discharge, $t$ is time in decimal years, $\beta_i$ are fitted
coefficients, and $\varepsilon$ is the error term. The 2[nd] and 3[rd] terms on the right represent time and
discharge effects, respectively, whereas the 4[th] and 5[th] terms collectively represent cyclical
seasonal effects. For a full description of this method, see Hirsch and De Cicco (2015). In this
work, WRTDS was applied to obtain the time series of estimated daily concentration for each
constituent at each site. The difference between observed concentration ($C_{obs}$) and estimated
concentration ($C_{est}$) was calculated in logarithmic space to obtain the concentration residuals,

$$residuals = ln(C_{obs}) - ln(C_{est}) \qquad (18)$$





For our data sets, histograms of concentration residuals (expressed in natural log concentration
units) are shown in **Figures S11-S14**. Compared with the original concentration data, these
model residuals are much more nearly normal and homoscedastic. Moreover, the model residuals
are less susceptible to the issues of temporal, seasonal, and discharge-drive variations than the
original concentrations. Therefore, the model residuals are more appropriate than the original
concentrations for $\beta$ estimation using the simulation framework adopted in this work.
The estimated $\beta$ values for the concentration residuals are summarized in **Figure 10**. Clearly,
the estimated $\beta$ varies considerably with the estimation method. In addition, the estimated $\beta$
varies with site and constituent (*i.e.*, TP, TN, or $NO_x$.) Our discussion below focuses on the
wavelet method (C2), because it is established above that this method performs better than the
other estimation methods under a wide range of gap conditions. We emphasize that it is beyond
our current scope to precisely quantify $\beta$ in these water-quality data sets, but our simulation
results presented above (**Section 3.2**) can be used as references to qualitatively evaluate the
reliability of C2 and/or other methods for these data sets.
For TN and TP concentration data at the Chesapeake River Input Monitoring sites (**Table 1**),
$\mu$ varies between 9.5 and 24.4, whereas $\lambda$ is ~1.0. Thus, the simulated gap scenario of NB($\mu = 14$,
$\lambda = 1$) can be used as a reasonable reference to assess methods' reliability (**Figure 8**). Based on
method C2, the estimated $\beta$ ranges between $\beta = 0.36$ and $\beta = 0.61$ for TN and between $\beta = 0.30$
and $\beta = 0.58$ for TP at these sites (**Figure 10**). For such ranges, the simulation results indicate
that method C2 tends to moderately under-estimate $\beta$ under this gap scenario (**Figure 8**), and
hence spectral slopes for TN and TP at these Chesapeake sites are likely slightly higher than
those presented above.
For $NO_x$ and TP concentration data at the Lake Erie and Ohio sites (**Table 1**), $\mu$ varies
between 0.06 and 0.22, whereas $\lambda$ is ~0.01. Thus, the simulated gap scenario of NB($\mu = 1$, $\lambda =$
0.01) can be used as a reasonable reference to assess the methods' reliability (**Figure 6**). For
such small $\lambda$ (*i.e.*, a few gaps that are very dissimilar from others), C2 is not reliable for $\beta$
estimation, as reflected by the generally positive bias in the simulation results. By contrast,
methods B1 (interpolation with global mean) and B8 (lowess with span 0.75) both perform quite
well under this gap scenario (**Figure 6**). These two methods provide almost identical $\beta$ estimates
for each site-constituent combination, ranging from $\beta = 1.0$ to $\beta = 1.5$ for $NO_x$ and from $\beta = 1.0$
to $\beta = 1.4$ for TP (**Figure 10**).



Overall, the above analysis of real water-quality data has illustrated the wide variability in $\beta$
estimates, with different choices of estimation methods yielding very different results. To our
knowledge, these water-quality data have not heretofore been analyzed in this context. As
illustrated above, our simulation experiments (**Section 3.2**) can be used as references to coarsely
evaluate the reliability of each method under specific gap scenarios, thereby considerably
narrowing the likely range of the estimated spectral slopes. Nonetheless, our results demonstrate
that the analyzed water-quality time series can exhibit strong fractal scaling, particularly at the
Lake Erie and Ohio tributary sites. Thus, an important implication is that researchers and
analysts should be cautious when applying standard statistical methods to identify temporal
trends in such water-quality data sets (Kirchner and Neal, 2013). In future work, one may
consider applying Bayesian statistical analysis or other approaches to more accurately quantify
the spectral slope and associated uncertainty for real water-quality data analysis. In addition, the
modeling framework presented herein (including both gap simulation and $\beta$ estimation) may be
extended to simulations of irregular time series that have prescribed spectral slopes and also
superimposed temporal trends, which can then be used to evaluate the validity of various
statistical methods for identifying trend and associated statistical significance.
**4. Conclusions**
River water-quality time series often exhibit fractal scaling behavior, which presents
challenges to the identification of deterministic trends. Because traditional estimation methods
are generally not applicable to irregularly sampled time series, we have examined two broad
types of estimation approaches and evaluated their performances against synthetic data with a
wide range of prescribed $\beta$ values and gap intervals representative of the sampling irregularity of
real water-quality data.
The results of this work suggest several important messages. First, the results remind us of
the risks in using interpolation for gap filling when examining auto-correlation, as the
interpolation methods consistently under-estimate or over-estimate $\beta$ under a wide range of
prescribed $\beta$ values and gap distributions. Second, the long-established Lomb-Scargle spectral
method also consistently under-estimates $\beta$. Its modified form, using the 5% lowest frequencies
for spectral slope estimation, has very poor precision, although the overall bias is small. Third,
the wavelet method, coupled with an aliasing filter, has the smallest bias and root-mean-squared



error among all methods for a wide range of prescribed $\beta$ values and gap distributions, except for
cases with small prescribed $\beta$ values (*i.e.*, close to white noise) or small $\lambda$ values (*i.e.*, very
skewed gap distributions). Thus, the wavelet method is recommended for estimating spectral
slope in irregular time series until improved methods are developed. In this regard, future
research should aim to develop an aliasing filter that is more applicable to irregular time series
with very skewed gap intervals. Finally, all methods' performances depend strongly on the
sampling irregularity in terms of both the skewness and mean of gap-interval lengths,
highlighting that the accuracy and precision of each method are data-specific.

Overall, these results provide new contributions in terms of better understanding and

quantification of the proposed methods' performances for estimating the strength of fractal
scaling in irregularly sampled water-quality data. In addition, the work has provided an
innovative and general approach for modeling sampling irregularity in water-quality records.
Moreover, this work has proposed and demonstrated a generalizable framework for data
simulation (with gaps) and $\beta$ estimation, which can be readily applied toward the evaluation of
other methods that are not covered in this work. More generally, the findings and approaches
may also be broadly applicable to irregularly sampled data in other scientific disciplines. Last but
not least, we note that accurate quantification of fractal scaling in irregular water-quality time
series remains an unresolved challenge for the hydrologic community and for many other
disciplines that must grapple with irregular sampling.
**Data Availability**
River monitoring data used in this study are available through the U.S. Geological Survey
National Water Information System (http://doi.org/10.5066/F7P55KJN) and the Heidelberg
University's National Center for Water Quality Research.
**Supporting Information**
Supporting information to this article is available online.
**Competing Interests**
The authors declare that they have no conflict of interest.





## Acknowledgements


Q. Zhang was supported by the Maryland Sea Grant through awards NA10OAR4170072 and
NA14OAR1470090 and by the Maryland Water Resources Research Center through a graduate
fellowship while he was a doctoral student at the Johns Hopkins University. C. Harman's contribution to
this work was supported by the National Science Foundation through grants CBET-1360415 and EAR-
1344664. We thank Bill Ball (Johns Hopkins University) and Bob Hirsch (U.S. Geological Survey) for
many useful discussions.

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




**Table 1.** Quantification of sampling irregularity for selected water-quality constituents at nine sites of the Chesapeake Bay River Input Monitoring program and six sites of the Lake Erie and Ohio tributary monitoring program. ($\mu$: mean parameter; $\lambda'$: shape parameter estimated using maximum likelihood; $\lambda$: shape parameter estimated using the direct approach (see **Section 2.2**). $\Delta t_{average}$: average gap interval; $N$: total number of samples.)

*I. Chesapeake Bay River Input Monitoring program*

| Site ID | River and station name | Drainage area (mi²) | Total nitrogen (TN) | | | | | Total phosphorus (TP) | | | | |
|---|---|---|---|---|---|---|---|---|---|---|---|---|
| | | | $\lambda$ | $\lambda'$ | $\mu$ | $\Delta t_{average}$ (days) | $N$ | $\lambda$ | $\lambda'$ | $\mu$ | $\Delta t_{average}$ (days) | $N$ |
| 01578310 | Susquehanna River at Conowingo, MD | 27100 | 0.8 | 1.1 | 13.5 | 14.5 | 876 | 0.8 | 1.0 | 13.4 | 14.4 | 881 |
| 01646580 | Potomac River at Chain Bridge, Washington D.C. | 11600 | 0.9 | 0.6 | 9.5 | 10.5 | 1385 | 1.1 | 1.0 | 24.4 | 25.4 | 579 |
| 02035000 | James River at Cartersville, VA | 6260 | 0.8 | 1.0 | 13.9 | 14.9 | 960 | 0.8 | 1.1 | 13.7 | 14.7 | 974 |
| 01668000 | Rappahannock River near Fredericksburg, VA | 1600 | 0.8 | 0.6 | 15.6 | 16.6 | 776 | 0.8 | 0.6 | 15.2 | 16.2 | 796 |
| 02041650 | Appomattox River at Matoaca, VA | 1340 | 0.8 | 0.8 | 15.1 | 16.1 | 798 | 0.8 | 0.8 | 14.9 | 15.9 | 810 |
| 01673000 | Pamunkey River near Hanover, VA | 1071 | 0.8 | 0.9 | 15.1 | 16.1 | 873 | 0.8 | 1.0 | 14.7 | 15.7 | 894 |
| 01674500 | Mattaponi River near Beulahville, VA | 601 | 0.7 | 0.9 | 14.3 | 15.3 | 810 | 0.8 | 0.9 | 14.2 | 15.2 | 820 |
| 01594440 | Patuxent River at Bowie, MD | 348 | 0.9 | 1.1 | 15.3 | 16.3 | 787 | 0.8 | 0.8 | 14.0 | 15.0 | 861 |
| 01491000 | Choptank River near Greensboro, MD | 113 | 1.2 | 1.5 | 19.6 | 20.6 | 680 | 1.1 | 1.0 | 20.5 | 21.5 | 690 |

*II. Lake Erie and Ohio tributary monitoring program*

| Site ID | River and station name | Drainage area (mi²) | Nitrate-plus-nitrite (NOx) | | | | | Total phosphorus (TP) | | | | |
|---|---|---|---|---|---|---|---|---|---|---|---|---|
| | | | $\lambda$ | $\lambda'$ | $\mu$ | $\Delta t_{average}$ (days) | $N$ | $\lambda$ | $\lambda'$ | $\mu$ | $\Delta t_{average}$ (days) | $N$ |
| 04193500 | Maumee River at Waterville, OH | 6330 | 0.005 | 0.0003 | 0.19 | 1.19 | 9101 | 0.005 | 0.0003 | 0.19 | 1.19 | 9101 |
| 04198000 | Sandusky River near Fremont, OH | 1253 | 0.01 | 0.003 | 0.22 | 1.22 | 9641 | 0.01 | 0.003 | 0.22 | 1.22 | 9655 |
| 04208000 | Cuyahoga River at Independence, OH | 708 | 0.007 | 0.006 | 0.13 | 1.13 | 7421 | 0.007 | 0.006 | 0.13 | 1.13 | 7426 |
| 04212100 | Grand River near Painesville, OH | 686 | 0.01 | 0.005 | 0.21 | 1.21 | 5023 | 0.01 | 0.005 | 0.22 | 1.22 | 4994 |
| 04197100 | Honey Creek at Melmore, OH | 149 | 0.007 | 0.005 | 0.06 | 1.06 | 9914 | 0.007 | 0.005 | 0.06 | 1.06 | 9914 |
| 04197170 | Rock Creek at Tiffin, OH | 34.6 | 0.007 | 0.008 | 0.06 | 1.06 | 8422 | 0.007 | 0.008 | 0.06 | 1.06 | 8440 |



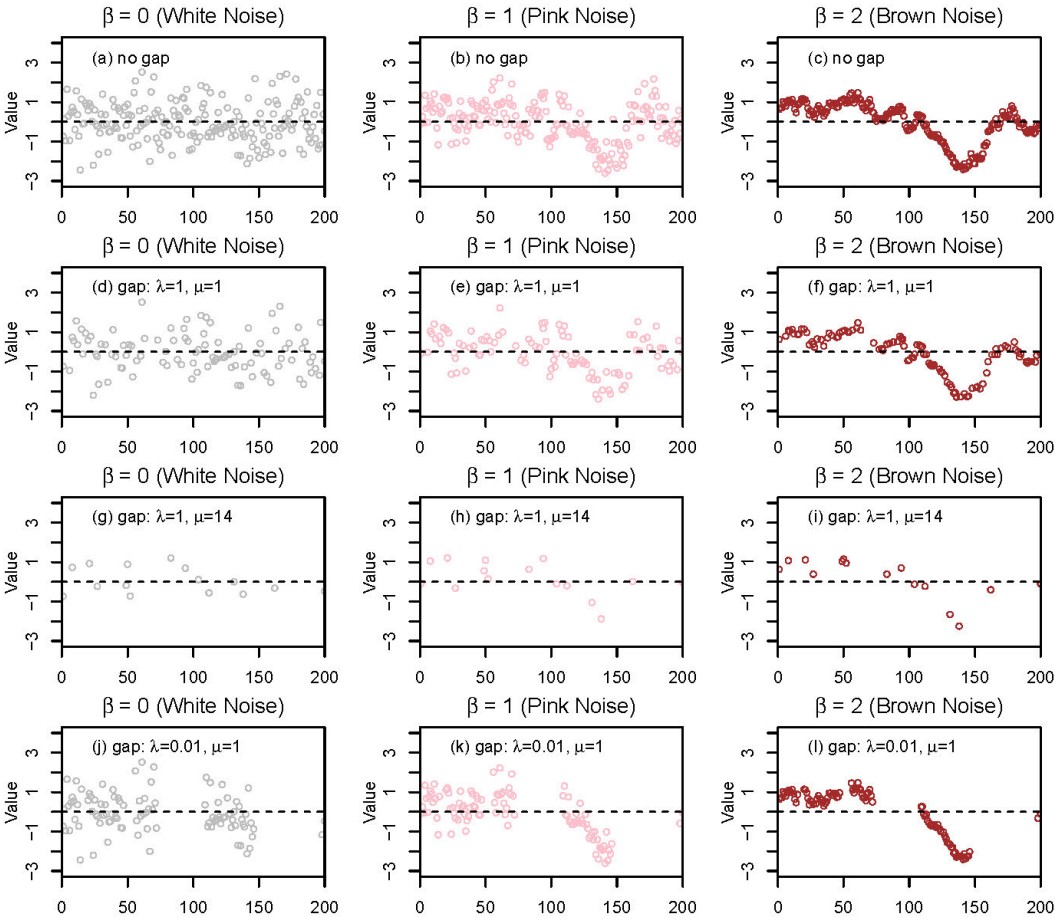

**Figure 1**. Synthetic time series with 200 time steps for three representative fractal scaling processes that correspond to white noise ($\beta = 0$), pink noise ($\beta = 1$), and Brown noise ($\beta = 2$). The 1st row shows the simulated time series without any gap. The three rows below show the same time series as in the 1st row but with data gaps that were simulated using three different negative binomial (NB) distributions, that is, 2nd row: NB($\lambda = 1, \mu = 1$); 3rd row: NB($\lambda = 1, \mu = 14$); 4th row: NB($\lambda = 0.01, \mu = 1$).





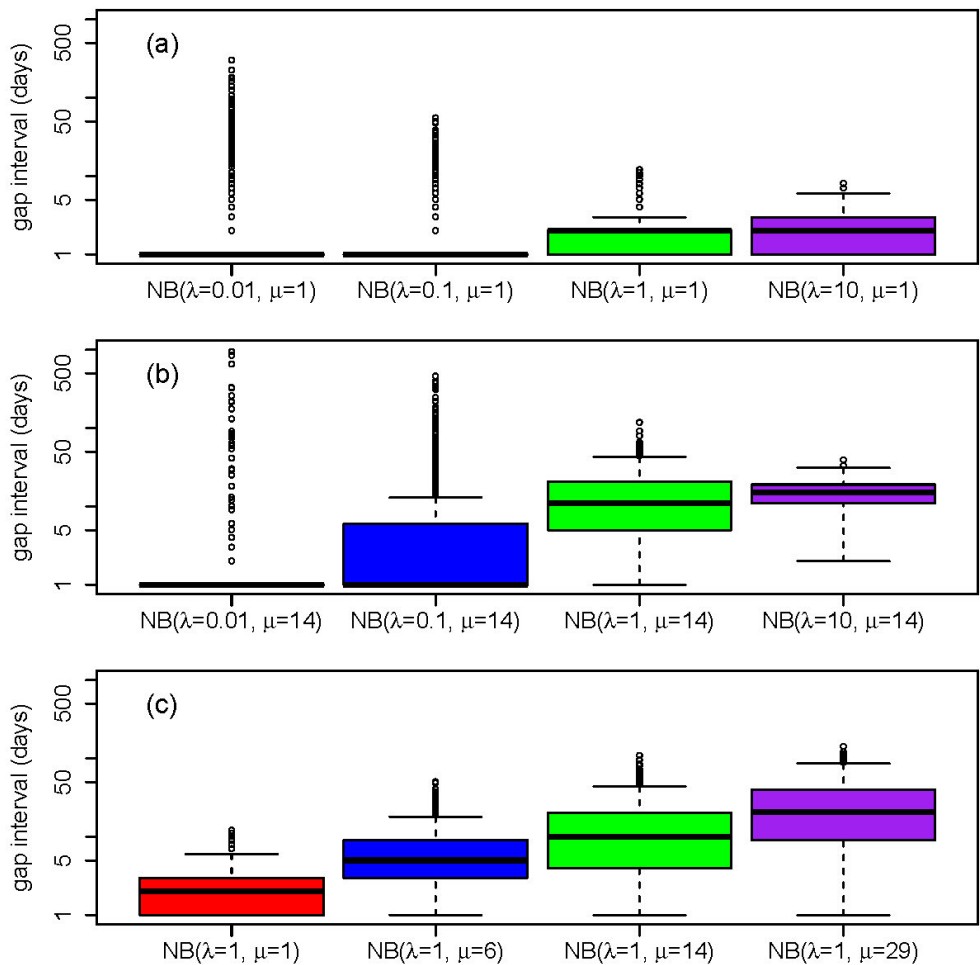

**Figure 2**. Examples of gap interval simulation using binomial distributions, NB (shape $\lambda$, mean $\mu$). Simulation parameters: $L = 9125$ days, $\Delta t_{nominal} = 1$ day. The three panels show simulation with fixed (a) $\mu = 1$, (b) $\mu = 14$, and (c) $\lambda = 1$. Note that $\Delta t_{average}/\Delta t_{nominal} = \mu + 1$.




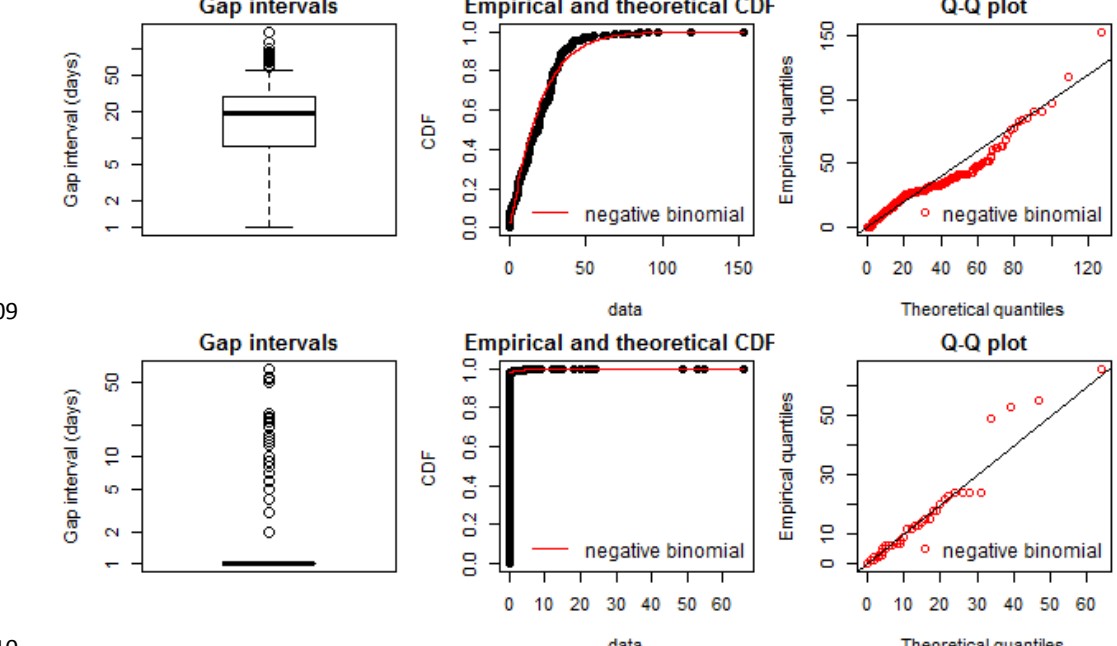


**Figure 3**. Examples of quantified sampling irregularity with negative binomial (NB)
distributions: total nitrogen in Choptank River (top) and total phosphorus in Cuyahoga River
(bottom). Theoretical CDF and quantiles are based on the fitted NB distributions. See **Table 1**
for estimated mean and shape parameters.



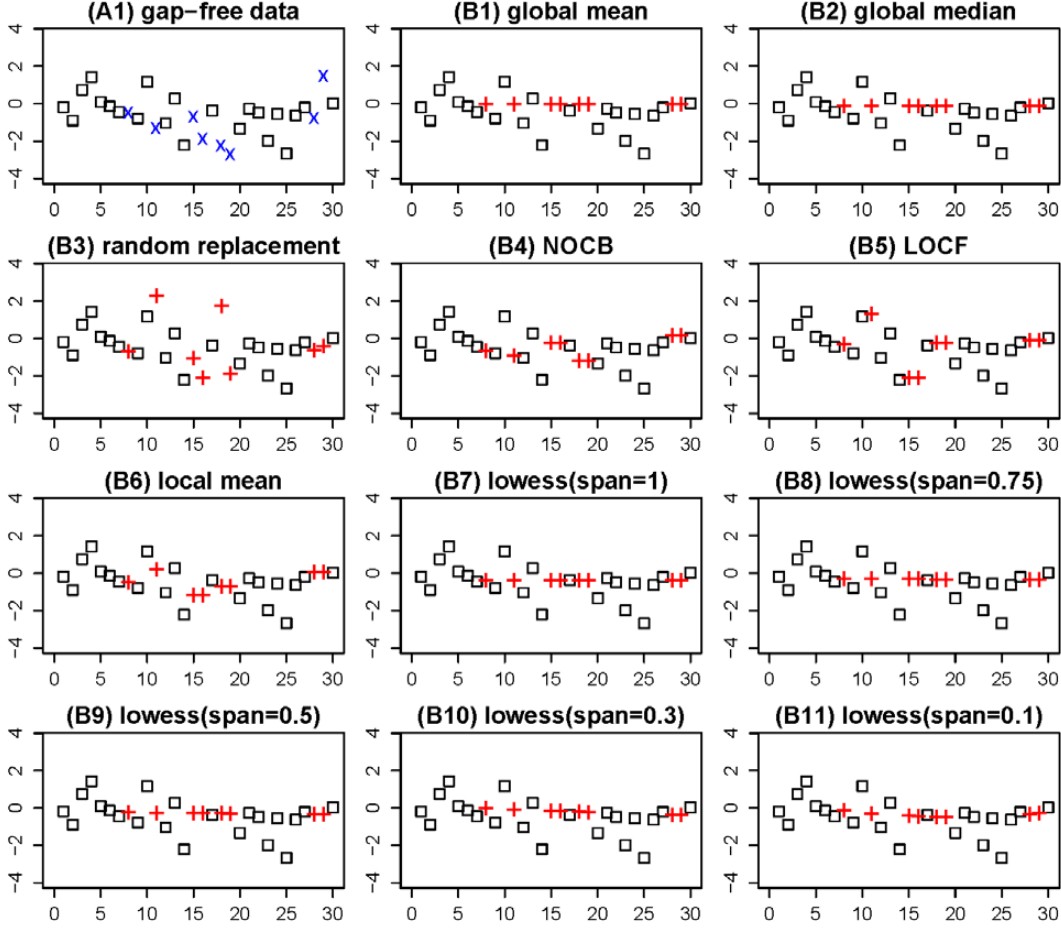


**Figure 4**. Illustration of the interpolation methods for gap filling. The gap-free data (A1) was

simulated with a series length of 500, with the first 30 data shown. (x: omitted data for gap filling;

+: interpolated data; NOCB: next observation carried backward; LOCF: last observation carried

forward; lowess: locally weighted scatterplot smoothing.)





720

**Figure 5**. Comparison of bias in estimated spectral slope in irregular data that are simulated with

prescribed $\beta = 1$ (30 replicates), series length of 9125, and gap intervals simulated with (a) NB ($\lambda$

$= 0.01$, $\mu = 1$), (b) NB ($\lambda = 0.1$, $\mu = 1$), (c) NB ($\lambda = 1$, $\mu = 1$), and (d) NB ($\lambda = 10$, $\mu = 1$). The blue

dashed lines indicate the true $\beta$ value.





**Figure 6.** Comparison of bias in estimated spectral slope in irregular data that are simulated with varying prescribed $\beta$ values (30 replicates), series length of 9125, and mean gap interval of 2 (*i.e.*, $\mu = 1$).





**Figure 7**. Comparison of root-mean-squared error (RMSE) in estimated spectral slope in irregular data that are simulated with varying prescribed $\beta$ values (30 replicates), series length of 9125, and mean gap interval of 2 (*i.e.*, $\mu = 1$).

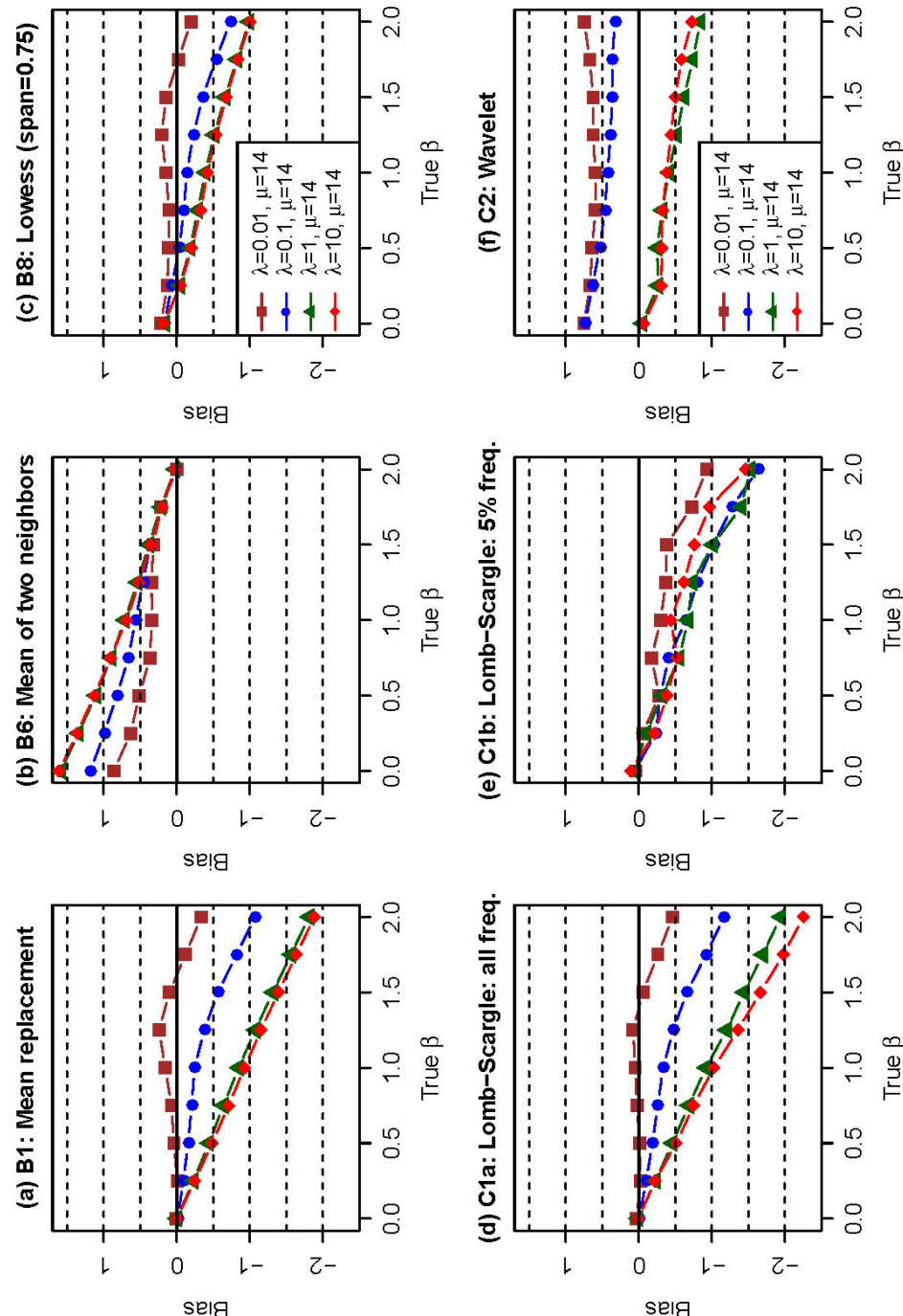

**Figure 8.** Comparison of bias in estimated spectral slope in irregular data that are simulated with varying prescribed $\beta$ values (30 replicates), series length of 9125, and mean gap interval of 15 (*i.e.*, $\mu = 14$).






**Figure 9.** Comparison of root-mean-squared error (RMSE) in estimated spectral slope in irregular data that are simulated with varying prescribed $\beta$ values (30 replicates), series length of 9125, and mean gap interval of 15 (*i.e.*, $\mu = 14$).




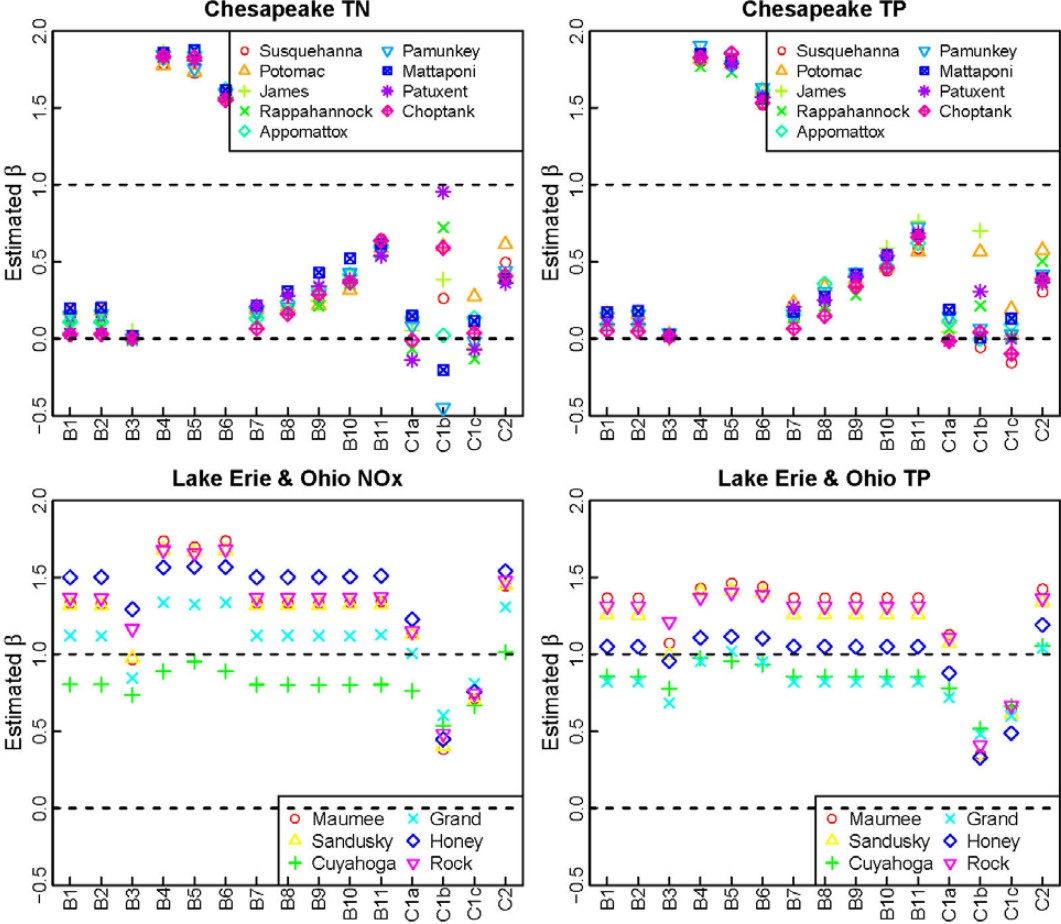

**Figure 10.** Quantification of spectral slope in real water-quality data from the two regional monitoring networks, as estimated using the set of examined methods. All estimations were performed on concentration residuals (in natural log concentration units) after accounting for effects of time, discharge, and season. The two dashed lines in each panel indicate white noise ($\beta = 0$) and pink (flicker) noise ($\beta = 1$), respectively. See **Table 1** for site and data details.