# Peer review of "Evaluation of statistical methods for quantifying fractal scaling in water quality time series with irregular sampling"

_Hydrology and Earth System Sciences, 2017_

## Referee Comment (RC1) · Anonymous Referee #1 · 24 Jul 2017

The study performs a comparative evaluation of statistical methods available in the literature using as benchmark test the quantification of fractal scaling in water quality time series with irregular sampling.

While lacking technical novelty, the manuscript is written in a sober, careful manner, aptly guiding the reader throughout the key arguments, methodologies and procedures, and presenting the results in a clear and concise manner.

This being said, I raise four fundamental concerns:

1) The paper is a pure statistical exercise. It would thus benefit from a physical interpretation of the methodological structure and results: namely, discussing possible

physical mechanisms responsible for the statistical signatures detected in the analysis, along with their physical consistency.

For instance, whether a trend is physically sustainable and realistic in terms of system energetics, what physical mechanisms sustain the power laws detected in the data, and what physically entails the fractal behaviour. Fractals and scaling are well understood in the physical sciences but the HESS readers will be happy to learn this in the hydrological context.

In doing so, the authors will be able to strengthen their arguments and diffuse concerns about whether there is any realism underlying the signatures detected in the analysis.

2) Monte Carlo simulations can be structured and tuned for essentially any purpose and to yield any outcome, relying on wise choices made in the methodological setup and the generating system, based on the researchers' understanding or conception of its behaviour.

If the methodological setup is entirely data-based, i.e. learn from some statistic or machine learning procedure derived from dataset records, there will always be a degree of case-specific empiricism that is not straightforwardly generalisable, unless there is a fundamental principle beneath. This again links to concern 1.

Therefore, it is important to thoroughly provide a solid background to all the assumptions supporting the choices made in the methodological setup and operation.

3) The paper does not introduce any methodological novelty. In fact, there is a vast literature on statistics of irregularly sampled series (also known as unevenly spaced time series). Therefore, I strongly encourage the authors to look into the literature outside of hydrology, e.g. in astrophysics, neurosciences, paleoclimatology, where they will find a rich diversity of sophisticated and long-proven methods that already tackle the same problems.

In doing so, the authors will necessarily tone down the false claims about novelty in

new methods and frameworks, when in reality the only novelty is the application of existing methods to hydrological case studies.

The key merit of the paper is essentially the comparative evaluation of well known statistical methods and their application to the hydrological sciences, namely relevant water quality issues.

As such, this is a purely applied paper and should be clearly presented as such. This brings me to the fourth concern.

4) There are no novel hydrological insights in the paper. While the statistical messages are useful (albeit not technically novel), it would be essential to bring out a substantial advance in the understanding of the hydrological and earth systems. After all, HESS is not merely a journal of applied statistics but rather one in which there should be something to be learnt in the functioning of the hydrological system.

---

## Referee Comment (RC2) · Anonymous Referee #2 · 25 Jul 2017

I find the manuscript is well-written and technically rigorous, with results that can be generalized beyond hydrologic time series. This manuscript tackles a challenging and highly relevant topic - the quantification of fractal scaling behavior for irregularly sampled data - and provides needed synthesis on the most promising methods to estimate this behavior. For these reasons, I recommend the manuscript be accepted subject to minor revisions.

I do, however, have a number of comments that would help improve clarity of the manuscript and emphasize the more practical aspects of this work.

Major comments:

1) Lines 127-129: It would be interesting to the reader and for understanding the important contribution of this work to detail the effects of non-normal data and persistence, seasonality, and the presence of long-term trends on the estimation of Beta.

2) Lines 264-265: It is noted that the results which demonstrate that the approach used in this manuscript to mimic the sampling irregularity performs well as compared to other simulation methods are not shown. I think these results are important to show, as this approach is what underlies the remainder of the analysis of the methods. This can be added to the supplementary material.

3) There are a large number of interpolation methods (n=11) presented here. I would argue that some of these methods are not very realistic in the context of what one would experience in terms interpolation for irregular samples. Unless the authors provide sound technical justification for each scenario, I would consider removing scenarios that would not generally be considered in standard practices (examples are scenarios B3, B4, and select a smaller subset of LOESS smoothing parameter values). This would also streamline the results and text.

4) Line 412: For Monte Carlo analysis, average values of the simulated parameter of interest are computed from sample sizes of 100 or more - not 30. Was this tested in your experiments?

Minor clarification comments:

1) Lines 3-4: Consider adding a phrase or sentence to explain why spectral slope is important to trend detection.

2) Line 15: Is the "modified form" being newly introduced here? Or does it already exist. Clarify.

3) line 38-39: The fact that ACF is summable seems a non-sequitor here. It is later that the connection is made to summability and the presence of fractal behavior. Perhaps it is not necessary to comment on the summability of the ACF?

4) lines 90-103: Moving this paragraph to the end of Section 1.1 would provide more immediate clarity as to the scope and value of this work.

5) Line 158: Consider the of the work "interpolating" instead of "modeling"

6) Section 2.1: Highly clever way to define sampling irregularity.

7) line 216 (and throughout): I do not think "gappy" is a word and chuckled at its appearance. Please replace with "irregularly-spaced".

8) line 237: Please be more specific in how you arrived at this equation for the shape parameter.

9) line 247 (as an example): Please add units to values provided in this section and throughout. This will help the reader follow the results and methods.

10) line 473: Hirsch and DeCicco (2015) is the reference to the user manual for WRTDS. The method itself is explained in Hirsch et al. (2010). I would cite the original paper.

Hirsch, R. M., Moyer, D. L. and Archfield, S. A. (2010), Weighted Regressions on Time, Discharge, and Season (WRTDS), with an Application to Chesapeake Bay River Inputs. JAWRA Journal of the American Water Resources Association, 46: 857–880. doi: 10.1111/j.1752-1688.2010.00482.x

---

## Referee Comment (RC3) · Anonymous Referee #1 · 26 Sep 2017

Thank you for your time and diligence in preparing your responses. Albeit the well articulated arguments, which deserved my full consideration, unfortunately they do not yet solve some of the fundamental scientific concerns that the paper raises.

Before discussing such concerns in detail, I would like to note that I have great consideration for the technical merits of the work undergone in the manuscript, and for the problems tackled by the authors in the present quest. Fractal scaling in irregular time series is a worthy venture with important relevance in many technical fields, and it has well-proven scientific merits in various disciplines as the reference list in point 5 of this report documents.

This work shall also become more relevant on a scientific standpoint once it is complemented by a sound scientific basis advancing the understanding of hydrologic functioning, namely on the fundamental principles that explain the fundamental nature of the detected signatures and the implemented approaches beyond the already presented statistical and geometric considerations.

In detail, this report discusses the following key concerns:

1) On the hydrological insights or lack thereof:

With all due respect, and albeit the arguments put forward by the authors, the paper is still fundamentally devoid of any hydrological insights. The well-known fractal scaling approaches discussed in the manuscript have purely descriptive merits and are inherently grounded on statistical geometry. The underlying physical understanding is thus entirely missing.

The ability of fractal scaling to elicit trends is again a purely descriptive merit, adding nothing to the fundamental understanding of hydrological functioning. To reach that understanding, one must fundamentally address the questions: Why? What are the process reasons behind an observed statistical and/or geometric signature? What is the physical meaning of a fractional scaling exponent? Of a fractional law? And what do these inform about the "hows" and "whys" in hydrology?

2) On the long history of fractal theories and scaling - centuries before hydrology:

The science underlying fractal analysis, geometry and scaling has a long scientific history before the empirical work of Hurst. While he contributed to the field from its empirical side, the fundamental contributions date back to the 17th century calculus. You may have learnt at the university about the many contributions of Leibniz: among many contributions, he introduced the first solid concepts of fractional exponents and laid the foundations of fractional calculus (aside from making fundamental contributions to mainstream calculus as well). We owe Leibniz - not Hurst - the fundamentals on

fractal scaling.

Soon these 17th century concepts were linked to physical laws and a new branch of analytical and statistical mechanics was born. Among the many illustrious users of fractional mechanics was Einstein, which provided systematic rigorous physical grounds to what predecessors had only described with geometry and statistics (the physics behind the random walks in Brownian motion). This is actually what landed him the Nobel prize in Physics (instead of his more famous contributions on relativity and quantum mechanics).

3) On the descriptive nature of the hydrological work - science still elusive

In its current state, hydrology is still an applied discipline, and there is nothing wrong about that in principle. The service that the discipline plays to society is undeniable and is ultimately why all of us gather here in this forum to advance the field for the benefit of all. While some important laws have been formulated, the discipline still lacks a consistent fundamental theory, and all theoretical formulations are imported from elsewhere (e.g. Darcy's law is just an import of the Ohm's law to hydrology).

Living off empiricisms and imports without any fundamental theoretical explanation may be good in statistical and engineering hydrology but will not lead us anywhere in real science. In fact, the inherent empiricism of many hydrological literature is understandable in an engineering setting where all that matters is to get some number that ensures portability of measured features from one scale to another for design and decision support. However, that brings no understanding about the real functioning of the hydrologic system.

The work presented in this manuscript has a good place in an applied statistics or engineering setting, for use by practitioners in engineering hydrology that have no time or scientific background to study, understand or build on the real scientific literature.

However, as a candidate for scientific paper, there is, at this point, no science to be

learnt in the present study. Venturing into approaches without complementing them with the supporting physical principles brings little benefit to hydrological understanding, since their descriptive aptitude is not accompanied by any physically related insight.

4) On the (ir)relevance of comparing various methods that in essence are more of the same

The different approaches compared in the study belong to the same class of methodological equivalence of naïve statistical geometry, and do not necessarily represent the best in the field. Therefore, comparing the analysed approaches is of little methodological relevance since it is not taking a comprehensive and useful up-to-date selection.

In other words, there little relevance in performing a comparison among different methods that are no longer up to date, and even less so when they belong to the same methodological class of equivalence. An illustrative view to get the idea: publishing a study comparing the various models of chariot transport will add nothing relevant in the age of the automobile, unless we are interested in the history of science and technology. At best, we should compare methods that are fundamentally different from each other rather than variants of the very same concept.

5) On the "onus of proof"

It is the duty of the authors to unequivocally demonstrate a significant degree of scientific innovation and novel hydrological insights. So far, from the manuscript and responses, that unequivocal proof is still missing.

While it is not the referee's duty to demonstrate the vacuity of the study - but rather the authors' duty to demonstrate its substance - I will gladly provide elements to help the authors find a wealth of literature on studies that cover the same problems and solutions discussed in the present study.

A careful and thorough literature review can aptly demonstrated that existing studies

actually perform in due terms what the present article claims was missing in the literature, thus effectively deflating the innovation claims.

For instance, the quantification of fractal scaling in irregularly sampled time series is a well studied problem and the experts are well aware about which methods perform better under which circumstances, therefore there is no gap in that area that would support any claim of innovation and relevance.

The references are found at the end of this report, at point 5.

4) On the ways forward to improve the manuscript:

The application of more comprehensive and effective fractal scaling techniques to your particular hydrological problems must be accompanied with real insights on hydrological functioning, rather than simply statistical description of results (e.g. a “trend” is hardly an insight, for it has descriptive but no explanatory power as noted above. Moreover, there is far more to fractal scaling than what is argued in the paper - again, the physics that would help elicit the “whys” in hydrologic functioning are missing).

In sum, my recommendation is to: a) bring real hydrological understanding rather than vague generic considerations and descriptive statements without quantitative physical reasoning to substantiate any explanatory claim; b) interpret the fractal metrics from a physical point of view that will shed light on the science behind the metric; c) provide if and how hydrologic series differ from others where fractal scaling over irregular series has already been extensively applied to justify separate publication.

At this stage, I leave the authors with some of the relevant literature references (apologies for non-uniform formatting):

5) REFERENCES:

A) Fractal approaches in irregular time series analysis:

Higuchi, T. (1988). Approach to an irregular time series on the basis of the fractal theory. Physica D: Nonlinear Phenomena. 31. 277-283. 10.1016/0167-2789(88)90081-4.

P. Paramanathan, R. Uthayakumar (2007): ICCIMA '07 Proc. Intl. Conf. on Computational Intelligence and Multimedia Applications (ICCIMA 2007), 02, 323-327

Peng CK, Havlin S, Stanley HE, Goldberger AL. Quantification of scaling exponents and crossover phenomena in nonstationary heartbeat time series. Chaos. 1995;5(1):82-7.

Stanley HE, Amaral LA, Goldberger AL, Havlin S, Ivanov PCh, Peng CK. Statistical physics and physiology: monofractal and multifractal approaches. Physica A. 1999 Aug 1;270(1-2):309-24.

Viswanathan GM1, Peng CK, Stanley HE, Goldberger AL. (1997): Phys Rev E Stat Phys Plasmas Fluids Relat Interdiscip Topics. 1997 Jan;55(1):845-9. Deviations from uniform power law scaling in nonstationary time series.

L. Telesca, V. Cuomo, V. Lapenna, M. Macchiato, C. Serio (1999): Detecting Stochastic Behaviour and Scaling Laws in Time Series of Geomagnetic Daily Means Pure appl. geophys. 156 (1999) 487–501, doi.org/10.1007/s000240050309.

Sergio Cerutti, Carlo Marchesi (2011): Advanced Methods of Biomedical Signal Processing. John Wiley & Sons.

M. Malik (1998): Clinical Guide to Cardiac Autonomic Tests. Springer Science & Business Media.

F Cervantes-De la Torre et al (2013): Fractal dimension algorithms and their application to time series associated with natural phenomena. J. Phys.: Conf. Ser. 475 012002.

B1) Analysis of irregularly sampled time series:

Roberto Vio et al (2000): PASP 112 74. https://doi.org/10.1086/316495

Piet M.T.Broersen (2005): Time series analysis for irregularly sampled data. IFAC Proc. Vol., 38, 1, 2005, 154-159 (http://www.sciencedirect.com/science/article/pii/S1474667016360384)

J.Durbin, S.J.Koopman (2012): Time Series Analysis by State Space Methods, 2nd edition, 2012. [See e.g. sec. 4.10, on statistical modeling under missing observations]

Adolf Mathias, Florian Grond, Ramon Guardans, Detlef Seese, Miguel Canela, Hans H. Diebner (2004): Algorithms for Spectral Analysis of Irregularly Sampled Time Series. Journal of Statistical Software. 10.18637/jss.v011.i02.

P Stoica, N Sandgren (2006): Spectral analysis of irregularly-sampled data: Paralleling the regularly-sampled data approaches. Digital Signal Processing, 2006, Elsevier.

Dilmaghani, Shabnam, et al. (2007): Harmonic analysis of environmental time series with missing data or irregular sample spacing. Environmental science & technology 41.20 (2007): 7030-7038.

Babu, Prabhu. Spectral analysis of nonuniformly sampled data and applications. Diss. Uppsala universitet, 2012.

B2) Analysis of unequally spaced data

JONES, R. H. (1985) Time series analysis with unequally spaced data. In Handbook of Statistics, Volume 5: Time Series in the Time Domain, ed. E. J. Hannan, P. R. Krishnaiah and M. M. Rao. North-Holland, Amsterdam.

Jones, R. (1986). Time Series Regression with Unequally Spaced Data. Journal of Applied Probability, 23, 89-98. doi:10.2307/3214345

C Hackman and T E Parker (1996): Noise analysis of unevenly spaced time series data. Metrologia 33 457

Eckner, Andreas. (2017). A Framework for the Analysis of Unevenly Spaced Time Series Data.

Doan TK, Haslett J, Parnell AC (2015): Joint inference of misaligned irregular time series with application to Greenland ice core data

Nieto-Barajas LE, Sinha T (2014): Bayesian interpolation of unequally spaced time series

Piet M T Broersen (2008): Time series models for spectral analysis of irregular data far beyond the mean data rate. Meas. Sci. Technol. 19 015103. https://doi.org/10.1088/0957-0233/19/1/015103

P.M.T. Broersen ; R. Bos (2006): Estimating time-series models from irregularly spaced data. IEEE Transactions on Instrumentation and Measurement

Piet M. T. Broersen, "Five Separate Bias Contributions in Time Series Models for Equidistantly Resampled Irregular Data", Instrumentation and Measurement IEEE Transactions on, vol. 58, pp. 1370-1379, 2009, ISSN 0018-9456.

Piet M.T. Broersen, "Bias Contributions in Time Series Models for Resampled Irregular Data", Instrumentation and Measurement Technology Conference Proceedings 2008. IMTC 2008. IEEE, pp. 882-889, 2008, ISSN 1091-5281.

Piet M. T. Broersen, "The Removal of Spurious Spectral Peaks From Autoregressive Models for Irregularly Sampled Data", Instrumentation and Measurement IEEE Transactions on, vol. 59, pp. 205-214, 2010, ISSN 0018-9456.

Piet M. T. Broersen, "Autoregressive Order Selection for Irregularly Sampled Data", Instrumentation and Measurement Technology Conference 2006. IMTC 2006. Proceedings of the IEEE, pp. 1004-1009, 2006, ISSN 1091-5281.

Piet M.T. Broersen, "Spectral Estimation from Irregularly Sampled Data for Frequencies Far Above the Mean Data Rate", Instrumentation and Measurement Technology Conference Proceedings 2007. IMTC 2007. IEEE, pp. 1-6, 2007, ISSN 1091-5281.

R. H. Jones, "Fitting multivariate models to unequally spaced data" in Time Series

Analysis of Irregularly Spaced Data, New York:Springer-Verlag, pp. 158-188, 1983.

E. K. Larsson, T. Söderström, "Identification of continuous-time AR processes from unevenly sampled data", Automatica, vol. 38, no. 4, pp. 709-718, Apr. 2002.

E. K. Larsson, E. G. Larsson, "The CRB for parameter estimation in irregularly sampled continuous-time ARMA systems", IEEE Signal Process. Lett., vol. 11, no. 2, pp. 197-200, Feb. 2002.

E. Lahalle, G. Fleury, A. Rivoira, "Continuous ARMA spectral estimation from irregularly sampled observations", Proc. IEEE/IMTC Conf., pp. 923-927, 2004.

S. de Waele, P. M. T. Broersen, "Error measures for resampled irregular data", IEEE Trans. Instrum. Meas., vol. 49, no. 2, pp. 216-222, Apr. 2000.

R. Bos, S. de Waele, P. M. T. Broersen, "Autoregressive spectral estimation by application of the Burg algorithm to irregularly sampled data", IEEE Trans. Instrum. Meas., vol. 51, no. 6, pp. 1289-1294, Dec. 2002.

R. H. Jones, "Maximum likelihood fitting of ARMA models to time series with missing observations", Technometrics, vol. 22, no. 3, pp. 389-395, 1980.

P. M. T. Broersen, S. de Waele, R. Bos, "Autoregressive spectral analysis when observations are missing", Automatica, vol. 40, no. 9, pp. 1495-1504, 2004.

P. M. T. Broersen, S. de Waele, R. Bos, "Application of autoregressive spectral analysis to missing data problems", IEEE Trans. Instrum. Meas., vol. 53, no. 4, pp. 981-986, Aug. 2004.

W. K. Harteveld, R. F. Mudde, H. E. A. van den Akker, "Estimation of turbulence power spectra for bubbly flows from laser Doppler anemometry signals", Chem. Eng. Sci., vol. 60, pp. 6160-6168, 2005.

P. M. T. Broersen; R. Bos (2006): Time-series analysis if data are randomly missing. IEEE Transactions on Instrumentation and Measurement, 55, 1, 79-84.

P. M. T. Broersen; S. de Waele; R. Bos (2003): Estimation of autoregressive spectra with randomly missing data Proc. 20th IEEE Instrumentation Technology Conf. (Cat. No.03CH37412), 2, 1154-1159.

R. Bos; S. de Waele; P. M. T. Broersen (2001): AR spectral estimation by application of the Burg algorithm to irregularly sampled data IMTC 2001. Proc. 18th IEEE Instrumentation and Measurement Technology Conf. Rediscovering Measurement in the Age of Informatics (Cat. No.01CH 37188), 2, 1208-1213.

C) Robust analysis of sparsely distributed data

Trendafilov, Nickolay; Kleinsteuber, Martin and Zou, Hui (2014). Sparse matrices in data analysis. Computational Statistics, 29(3-4) pp. 403–405.

Pires, C.A.L.; Perdigão, R.A.P. (2013): Minimum Mutual Information and Non-Gaussianity through the Maximum Entropy Method: Estimation from Finite Samples. Entropy 2013, 15, 721-752.

Peter Melchior, Andy D. Goulding (2016): Filling the gaps: Gaussian mixture models from noisy, truncated or incomplete samples. arXiv:1611.05806v1

Testa T., et al. (2016): Sparse representation of signals: from astrophysics to real-time data analysis for fusion plasmas and system optimization analysis for ITER and TCV. Plasma Physics and Controlled Fusion, Volume 58, Number 12. [review paper]

D) Astrophysical studies

Feigelson, E. D. & Babu, G. J. (1992): Improving the Statistical Methodology of Astronomical Data Analysis. Astronomical Data Analysis Software and Systems I, A.S.P. Conference Series, Vol. 25, 1992, Diana M. Worrall, Chris Biemesderfer, and Jeannette Barnes, eds., p. 237.

Bourguignon, S. and Carfantan, H. (2015): Line Spectra Estimation for Irregularly Sampled Signals in Astrophysics, in Regularization and Bayesian Methods for Inverse Problems in Signal and Image Processing (eds J.-F. Giovannelli and J. Idier), John Wiley & Sons, Inc., Hoboken, NJ, USA. doi: 10.1002/9781118827253.ch6

Jean-Luc Starck (2016): Sparsity and inverse problems in astrophysics. J. Phys.: Conf. Ser. 699 012010

E) Geo/environmental studies

K. Rehfeld and J. Kurths (2014): Similarity estimators for irregular and age-uncertain time series. Clim. Past, 10, 107–122, 2014. doi:10.5194/cp-10-107-2014

Björg Ólafsdóttir, Kristín; Schulz, Michael; Mudelsee, Manfred (2016): REDFIT-X: Cross-spectral analysis of unevenly spaced paleoclimate time series Computers and Geosciences, Volume 91, p. 11-18. 10.1016/j.cageo.2016.03.001

David J. Thomson: Time–Series Analysis of Paleoclimate Data. Encyclopedia of Paleoclimatology and Ancient Environments Part of the series Encyclopedia of Earth Sciences Series pp 949-959

I hope this helps.

Best wishes.

---

## Author Response (AR1)

**Response to Editor Comments**
Authors' responses inserted as blue text.

As I know both reviewers personally well and consider them as very much competent to judge your work, I decided to thoroughly and carefully review your manuscript to make up my mind in an as much as possible independent manner. As your manuscript is technically demanding, this took a little while. After this effort I am fully convinced that your work is very valuable for the field of hydrology, and yet I think needs moderate revisions before it can be accepted.

Response: Thank you very much for your thoughtful handling of this manuscript and for the additional efforts and time that you invested in as an additional reviewer.

Within this process I encourage you to address the recommendations of Reviewer 2 as outlined in your response.

Response: Thanks. We have thoroughly revised the manuscript based on our prior response to Reviewer 2.

While Reviewer 1 did, although she/he recommended major revisions, did not come up with a single major point that can be addressed, I leave it to you whether you consult a selection of the literature sources he/she provided or not.

Response: We have chosen not to modify the paper after Reviewer 1's comments. As we expressed in our prior responses to Review 1, we believe this is a useful contribution to the HESS community. We are very pleased to learn that both Reviewer 2 and the Editor have seen the merits of the manuscript.

Finally, although I am not an expert in time series analysis, I wonder about the following points:

1. In spatial statistics/geostatistics it is quite common to estimate the spatial covariance in a data set which is irregularly sampled in space by means of the semi-variogram. This implies to pool data into irregularly spaced lag classes and to fit the best suited theoretical variogram function, which of course might be also a power law. It is also standard to fill gaps between the sampling locations by interpolations methods that preserve the covariance structure for instance by means of ordinary kriging or external drift kriging, which may account for a drift in the expectation values (see next comment). I wonder whether the same procedure is not applicable to irregularly sampled time series, both to estimate the autocorrelation function and to fill gaps in a manner that preserves the autocorrelation?

Response: Great comment. While we think it is beyond the scope of our current manuscript, we agree that in future efforts it is useful to develop approaches to estimate autocorrelation (in the time domain) and then use that to fill the gaps in monitoring records. This kind of research can really improve our capability to build statistical models for estimating constituent concentrations and loads for days without samples. Unfortunately, this has been only well understood for simple autocorrelation models such as the autoregressive lag-1 model (AR[1]), which is a short-term memory model. But as we discuss in Section 1.1, water quality data generally exhibit long-term memory. Therefore, it remains a real challenge and also a critical need to "fill gaps in manner that preserves the autocorrelation" for long-memory time series.

Moreover, we want to add that since the auto-covariance is the Fourier transform of the spectrum, estimating the autocorrelation to infill the data before estimating the spectrum is really an example of 'begging the question' (in the original meaning of that idiom) in the sense of assuming a result in order to justify the conclusion.

2. The term water quality data seem a little imprecise. Do you refer to concentrations of solutes in stream flow (C) or to total loads (Q*C) in stream flow. Only the latter reflects mass conservation, and concentration data might exhibit quite a bit of pseudo-dynamics and pseudo seasonality due to simple dilution effects and seasonality in discharge (as you pointed out). In this context there is also quite a bit of discussion in the geostatistical literature whether data sets which include an external drift (a deterministic dependence of their mean on an external variable such as air temperature and or rainfall and topographic elevation) or in your case a flow dependence of concentration shall be detrended before estimating the covariance or not. I personally think it is necessary to remove a drive to avoid mixing of deterministic and stochastic variability – and there is evidence that a variogram analysis without the removal of a trend leads to a positive bias in the range and the sill of the variogram (e.g. Zimmermann et al. 2008).

Response: Thank you for this note on "external drift." We agree and that is why we discussed the complexities in concentration data with respect to trend, discharge, and season effects in the Introduction (see lines 139-153 in Section 1.3) and why we applied the WRTDS method to the real concentration data to remove such effects prior to the fractal scaling estimation (see Section 3.3).

3. The definition of delta t average seems inconsistent with example 2 in line 199.
Response: We double checked the definition and found the example consistent with its definition. For example 2, delta t average = (2+1+4+3)/4 = 2.5 hr, which also equals to L/(N-1) = 10/(5-1) = 2.5 hr.

4. Please flip the numbering of figure 1 and 2.
Response: Please note that Figure 1 appears first – below equation (8) in Section 1.2, so it is not needed to flip the two figures.

**Response to Anonymous Referee #2**
Authors' responses inserted as blue text.

I find the manuscript is well-written and technically rigorous, with results that can be generalized beyond hydrologic time series. This manuscript tackles a challenging and highly relevant topic - the quantification of fractal scaling behavior for irregularly sampled data - and provides needed synthesis on the most promising methods to estimate this behavior. For these reasons, I recommend the manuscript be accepted subject to minor revisions.
Response: Thank you for these comments.

I do, however, have a number of comments that would help improve clarity of the manuscript and emphasize the more practical aspects of this work.

Major comments:

1) Lines 127-129: It would be interesting to the reader and for understanding the important contribution of this work to detail the effects of non-normal data and persistence, seasonality, and the presence of long-term trends on the estimation of Beta.
Response: We agree this would be potentially useful to do, but it is simply beyond the scope of the paper -- new modeling experiments for each of these effects would multiply the length and complexity of the paper by large factors. We recognize this would be an important area for future research, which we explicitly put in the Section 3.3 (lines 459-464): "*such real data are typically much more complex than our simulated time series, because of (1) strong deviations from normal distributions and (2) effects of flow-dependence, seasonality, and temporal trend (Hirsch et al., 1991; Helsel and Hirsch, 2002). In this regard, future research may simulate time series with these important characteristics and evaluate the performance of various estimation approaches, perhaps following the modeling framework described herein.*"

2) Lines 264-265: It is noted that the results which demonstrate that the approach used in this manuscript to mimic the sampling irregularity performs well as compared to other simulation methods are not shown. I think these results are important to show, as this approach is what underlies the remainder of the analysis of the methods. This can be added to the supplementary material.
Response: Thank you for this suggestion. We now provide these simulation results in the supplementary material, see *Section S1*.

3) There are a large number of interpolation methods (n=11) presented here. I would argue that some of these methods are not very realistic in the context of what one would experience in terms interpolation for irregular samples. Unless the authors provide sound technical justification for each scenario, I would consider removing scenarios that would not generally be considered in standard practices (examples are scenarios B3, B4, and select a smaller subset of LOESS smoothing parameter values). This would also streamline the results and text.

Response: The interpolation methods were selected not only on the basis of their frequency of use, but to ensure a certain degree of completeness -- we felt it would ultimately be more useful to include obvious variations of common methods than to exclude one that someone might have been considering and looked to our paper for guidance. Although methods B3-B5 seem not plausible, they have been discussed and used in the scientific literature (e.g., Blankers et al. 2010; Graham 2009). Furthermore, some R packages have been developed (e.g., "na.locf"), making these methods readily available to general users without any prior knowledge on the methods' performance. Therefore, we think it is worthwhile to keep the results and discussion of these methods' performance, which is a useful contribution to the topic of "missing data analysis." Furthermore, removing one or two methods would do little to shorten or simplify the paper at this stage, so we have chosen not to so.

4) Line 412: For Monte Carlo analysis, average values of the simulated parameter of interest are computed from sample sizes of 100 or more - not 30. Was this tested in your experiments?

Response: We used 30 samples because it was quite sufficient to constrain estimates of the average in most cases. Standard error of the mean beta for most methods is much smaller than the variation between methods. Nonetheless, to follow common practice, we have adopted this suggestions to run the simulation 100 times and we have revised all relevant figures and text accordingly (including Figures 5-9 and Figures S3-S12.

Minor clarification comments:

1) Lines 3-4: Consider adding a phrase or sentence to explain why spectral slope is important to trend detection.

Response: We agree and have added the phrase "to avoid false inference on the statistical significance of trends" at the end of this sentence.

2) Line 15: Is the "modified form" being newly introduced here? Or does it already exist. Clarify.

Response: It is a published method, see method C1b in Section 3.1 where it is introduced. For clarity, we have revised "a modified form" to "a previously-published modified form."

3) line 38-39: The fact that ACF is summable seems a non-sequitor here. It is later that the connection is made to summability and the presence of fractal behavior. Perhaps it is not necessary to comment on the summability of the ACF?

Response: Agreed. We have deleted the comment on "summability".

4) lines 90-103: Moving this paragraph to the end of Section 1.1 would provide more immediate clarity as to the scope and value of this work.

Response: Thanks. This paragraph has been moved to the suggested location, i.e., Section 1.1.

5) Line 158: Consider the of the work "interpolating" instead of "modeling"
Response: We have decided to keep the use of "modeling."

6) Section 2.1: Highly clever way to define sampling irregularity.
Response: Thank you.

7) line 216 (and throughout): I do not think "gappy" is a word and chuckled at its appearance. Please replace with "irregularly-spaced".
Response: Thank you. We have replaced the word "gappy" with "irregularly-spaced" throughout the manuscript.

8) line 237: Please be more specific in how you arrived at this equation for the shape parameter.
Response: As the line states, $\lambda = \mu^2/[\mathrm{var}(\Delta t^*) - \mu] = (\mathrm{mean}(\Delta t^*) - 1)^2/[\mathrm{var}(\Delta t^*) - \mathrm{mean}(\Delta t^*) + 1]$. The first equality represents a well-established property for the negative binomial distribution. The second equality is achieved through the substitution of $\mu$ by "$\mathrm{mean}(\Delta t^*) - 1$". We think it is already clear and hence further modification is not necessary.

9) line 247 (as an example): Please add units to values provided in this section and throughout. This will help the reader follow the results and methods.
Response: Units are provided for $\Delta t_{average}$ (days). Note that $\lambda$ and $\mu$ are fitted (negative binomial distribution) parameters for the *non-dimensionalized* time series ($\Delta t^*$) -- see Section 2.1 -- therefore these two variables do not have units.

10) line 473: Hirsch and DeCicco (2015) is the reference to the user manual for WRTDS. The method itself is explained in Hirsch et al. (2010). I would cite the original paper.
Hirsch, R. M., Moyer, D. L. and Archfield, S. A. (2010), Weighted Regressions on Time, Discharge, and Season (WRTDS), with an Application to Chesapeake Bay River Inputs. JAWRA Journal of the American Water Resources Association, 46: 857–880. doi: 10.1111/j.1752-1688.2010.00482.x
Response: Thanks. We have corrected this citation to Hirsch et al. (2010).

References Cited
Blankers, M., Koeter, M. W., & Schippers, G. M. (2010). Missing data approaches in eHealth research: simulation study and a tutorial for nonmathematically inclined researchers. Journal of medical Internet research, 12(5).
Graham, J. W. (2009). Missing data analysis: Making it work in the real world. Annual review of psychology, 60, 549-576.

Authors' responses inserted as blue text.

The study performs a comparative evaluation of statistical methods available in the literature using as benchmark test the quantification of fractal scaling in water quality time series with irregular sampling. While lacking technical novelty, the manuscript is written in a sober, careful manner, aptly guiding the reader throughout the key arguments, methodologies and procedures, and presenting the results in a clear and concise manner.

Response: While we thank the reader for the comments about the clarity of our presentation, and for the helpful comments the reviewer has made, we disagree with the comment about the lack of technical novelty, which is repeated elsewhere below. We respond to it there.

This being said, I raise four fundamental concerns:

1) The paper is a pure statistical exercise. It would thus benefit from a physical interpretation of the methodological structure and results: namely, discussing possible physical mechanisms responsible for the statistical signatures detected in the analysis, along with their physical consistency. For instance, whether a trend is physically sustainable and realistic in terms of system energetics, what physical mechanisms sustain the power laws detected in the data, and what physically entails the fractal behaviour. Fractals and scaling are well understood in the physical sciences but the HESS readers will be happy to learn this in the hydrological context. In doing so, the authors will be able to strengthen their arguments and diffuse concerns about whether there is any realism underlying the signatures detected in the analysis.

Response: While we appreciate the value of a physical interpretation, we will not alter the manuscript in response to this comment for the following reasons:

I. We disagree that this is a 'statistical exercise' (which we take to imply that it provides no actual illumination). As its title reflects, the manuscript focuses on the comparison of various statistical methods for quantifying fractal scaling. These methods are actually used in the literature, and we believe that it is of value to know if they actually work or not.

II. Sections 1.1 and 1.2 in the Introduction provide background information with some reference to the physical origins, and providing many citations to literature that provides the insights the reviewer is looking for. In fact, significant sections of the foundational literature on fractals and scaling were developed within the hydrology community and later picked up more widely (the Hurst exponent is an example).

III. The physical interpretations of fractal time series are many and varied, depending on the context. Those interpretations are already available in the literature, and our purpose is neither to add to them, nor to review them. Instead our purpose is to conduct benchmark tests to determine whether some widely used techniques for inferring fractal scaling are reliable or not.

2) Monte Carlo simulations can be structured and tuned for essentially any purpose and to yield any outcome, relying on wise choices made in the methodological setup and the generating system, based on the researchers' understanding or conception of its behaviour. If the methodological setup is entirely data-based, i.e. learn from some statistic or machine learning procedure derived from dataset records, there will always be a degree of case-specific empiricism that is not straightforwardly generalisable, unless there is a fundamental principle beneath. This again links to concern 1. Therefore, it is important to thoroughly provide a solid background to all the assumptions supporting the choices made in the methodological setup and operation.

Response: We believe we have provided sufficient reasoning for the assumptions adopted, but welcome comments on any specific area that is unclear or has been omitted.

3) The paper does not introduce any methodological novelty. In fact, there is a vast literature on statistics of irregularly sampled series (also known as unevenly spaced time series). Therefore, I strongly encourage the authors to look into the literature outside of hydrology, e.g. in astrophysics, neurosciences, paleoclimatology, where they will find a rich diversity of sophisticated and long-proven methods that already tackle the same problems. In doing so, the authors will necessarily tone down the false claims about novelty in new methods and frameworks, when in reality the only novelty is the application of existing methods to hydrological case studies. The key merit of the paper is essentially the comparative evaluation of well known statistical methods and their application to the hydrological sciences, namely relevant water quality issues. As such, this is a purely applied paper and should be clearly presented as such. This brings me to the fourth concern.

Response: We agree with the reviewer's comment that "the paper is essentially the comparative evaluation of well known statistical methods and their application to the hydrological sciences, namely relevant water quality issues." This is precisely stated in Section 1.3 where we define the scope of the work -- see the 2nd last paragraph in that section. But we stress that we never claimed that our work is about developing "new methods". Rather, it is stated clearly in several locations of the paper that this work is about the evaluation of existing statistical methods.

In addition, we disagree with the reviewer on the point that there are "a rich diversity of sophisticated and long-proven methods that already tackle the same problems". Many existing methods do not apply to irregularly sampled data and hence can not be used. Others have been widely used, but have not been rigorously tested. The Lomb-Scargle spectral method is well established, but has known weaknesses (as discussed in the paper and elsewhere, see Montanari et al. 1999). If the reviewer is aware of other works that solve the problem addressed by our work, we again encourage the reviewer to provide citations.

Regarding the novelty (or contribution) of this work, we are not aware of any other papers that perform a similar comparative analysis of these methods, let alone one that is tailored to the needs of the hydrology and earth science community. The reviewer provides no evidence or citations to back up the claim that our study is not novel. If the reviewer is aware of any other such studies we would encourage the reviewer to provide them. One of us has been working on this problem for over 20 years, and in that time has seen no published work that is similar to ours.

Studies that review, compare and critically evaluate available methods are valuable contributions to the scientific literature. They are, in our opinion, useful checks on a proliferation of divergent methods that threatens to generate (at best) incomparable and (at worst) inaccurate observations of physical phenomena.

Our contribution in this regard is explicitly summarized in the last paragraph of the paper, which is copied below: "*Overall, these results provide new contributions in terms of better understanding and quantification of the proposed methods' performances for estimating the strength of fractal scaling in irregularly sampled water-quality data. In addition, the work has provided an innovative and general approach for modeling sampling irregularity in water-quality records. Moreover, this work has proposed and demonstrated a generalizable framework for data simulation (with gaps) and β estimation, which can be readily applied toward the evaluation of other methods that are not covered in this work. More generally, the findings and approaches may also be broadly applicable to irregularly sampled data in other scientific disciplines. Last but not least, we note that accurate quantification of fractal scaling in irregular water-quality time series remains an unresolved challenge for the hydrologic community and for many other disciplines that must grapple with irregular sampling.*"

4) There are no novel hydrological insights in the paper. While the statistical messages are useful (albeit not technically novel), it would be essential to bring out a substantial advance in the understanding of the hydrological and earth systems. After all, HESS is not merely a journal of applied statistics but rather one in which there should be something to be learnt in the functioning of the hydrological system.
Response: We disagree with the reviewer that this work does not provide contributions to HESS in terms of understanding of the hydrological and earth systems. As noted in Section 1.3 "Motivations and Objectives of this Work," the quantification of fractal scaling has important implications for detecting trends in water quality time series, but there is a large gap with respect to what methods are appropriate (or applicable) for quantifying fractal scaling in irregularly sampled water quality time series. By dealing with this issue, this work is highly relevant to the hydrological community.

References Cited

[revised manuscript text omitted]

---

## Author Response (AR2)

**Response to Reviewer Comments**

Authors' responses inserted as blue text.

I find that the authors have revised their manuscript to satisfy the reviewer comments and I recommend that the paper be accepted in its current form, subject to one minor comment. The phrase added to the abstract (p. 1, lines 2-3) to address one of the reviewer comments now makes the sentence confusing. I see two separate issues in the determining of trends when fractal scaling is present: 1) the presence of fractal scaling has the potential to lead to false inference about the statistical significance of trends, and 2) the abundance of irregularly spaced data in water-quality monitoring networks complicates the ability to quantify fractal scaling and identify trends. Please revise the abstract to reflect these two points. I think that will help improve the readability of the abstract especially with respect to lines 2-3.

Response: Thank you very much for your review of this manuscript. We are glad to hear that our revisions have improved the manuscript. We have revised the manuscript based on your comment. Please see the revised sentence in Abstract. For your convenience, this revised sentence is pasted below:

*Fractal scaling presents challenges to the identification of deterministic trends because (1) fractal scaling has the potential to lead to false inference about the statistical significance of trends and (2) the abundance of irregularly spaced data in water quality monitoring networks complicates efforts to quantify fractal scaling.*

**Notes to Editor**

Dear Editor,

Thank you again for handling this manuscript. In this revision, we have revised the manuscript based on the reviewer's suggestion.

In addition, we have also proofread the manuscript again and made a few grammatical revisions. These changes are marked in the "tracked change' version.

We hope that you will find it suitable for publication now.

Sincerely,

Qian

**Evaluation of statistical methods for quantifying fractal scaling in water quality time series with irregular sampling**

Qian Zhang[1], Ciaran J. Harman[2], James W. Kirchner[3,4,5]

[1] University of Maryland Center for Environmental Science at the US Environmental Protection Agency Chesapeake Bay Program Office, 410 Severn Avenue, Suite 112, Annapolis, MD 21403 (formerly, Department of Geography and Environmental Engineering, Johns Hopkins University, 3400 North Charles Street, Baltimore, Maryland 21218)

[2] Department of Environmental Health and Engineering, Johns Hopkins University, 3400 North Charles Street, Baltimore, Maryland 21218

[3] Department of Environmental System Sciences, ETH Zurich, Universitätstrasse 16, CH-8092 Zurich, Switzerland

[4] Swiss Federal Research Institute WSL, Zürcherstrasse 111, CH-8903 Birmensdorf, Switzerland

[5] Department of Earth and Planetary Science, University of California, Berkeley, California 94720

*Correspondence to*: Qian Zhang (qzhang@chesapeakebay.net)

**Abstract.** River water-quality time series often exhibit fractal scaling, which here refers to autocorrelation that decays as a power law over some range of scales. Fractal scaling presents challenges to the identification of deterministic trends because (1) fractal scaling has the potential to lead to false inference about the statistical significance of trends and (2) the abundance of irregularly spaced data in water quality monitoring networks complicates efforts to quantify fractal scaling. to avoid false inference on the statistical significance of trends, In the latter regard, but tTraditional 
[revised manuscript text omitted]